

# Uncertainty in Satellite estimate of Global Mean Sea Level changes, trend and acceleration

Michaël Ablain[1], Benoit Meyssignac[2], Lionel Zawadzki[1], Rémi Jugier[1], Aurélien Ribes[3], Anny Cazenave[2], Nicolas Picot [4]

[1] Collecte Localisation Satellite (CLS), Ramonville Saint-Agne, 31520, France
[2] LEGOS, CNES, CNRS, IRD, Université Paul Sabatier, Toulouse, France
[3] CNRM, Université Paul Sabatier, Météo France, CNRS, Toulouse, France
[4] Centre National d'Etudes Spatiales (CNES), Toulouse, 31400, France

*Correspondence to*: Michaël Ablain (ablain.michael1@gmail.com)

**Abstract.** Satellite altimetry missions now provide more than 25 years of accurate, continuous and quasi-global measurements of sea level along the reference ground track of TOPEX/Poseidon. These measurements are used by different groups to build the Global Mean Sea Level (GMSL) record, an essential climate change indicator. Estimating a realistic uncertainty of the GMSL record is of crucial importance for climate studies such

as estimating precisely the current rate and acceleration of sea level, analyzing the closure of the sea level budget, understanding the causes of sea level rise, detecting and attributing the response of sea level to anthropogenic activity, or estimating the Earth energy imbalance. (Ablain et al., 2015) estimated the uncertainty of the GMSL trend over the period 1993-2014 by thoroughly analyzing the error budget of the satellite altimeters and showed that it amounts to ±0.5 mm/yr (90% confidence level). In this study, we extend (Ablain et al., 2015)

analysis by providing a comprehensive description of the uncertainties in the satellite GMSL record. We analyse 25 years of satellite altimetry data and estimate for the first time the error variance-covariance matrix for the GMSL record with a time resolution of 10 days. Three types of errors are modelled (drifts, biases, noise) and combined together to derive a realistic estimate of the GMSL error variance-covariance matrix. From the error variance-covariance matrix we derive a 90% confidence envelop of the GMSL record on a 10-day basis. Then we

use a least square approach and the error variance-covariance matrix to estimate the GMSL trend and acceleration uncertainties over any time periods of 5 years and longer in between October 1992 and December 2017. Over 1993-2017 we find a GMSL trend of 3.35±0.4 mm/yr within a 90% Confidence Level (CL) and a GMSL acceleration of 0.12 ±0.07 mm/yr² (90% CL). This is in agreement (within error bars) with previous studies. The full GMSL error variance-covariance matrix is freely available online: https://doi.org/10.17882/58344 (Ablain

et al., 2018).



## 1 Introduction

Sea level change is a key indicator of global climate change which integrates changes in several components of the climate system in response to anthropogenic and natural climate variability. Since October 1992, sea level variations have been routinely measured by twelve high-precision altimeter satellites providing more than 25 years of continuous measurements. The altimeter Global Mean Sea Level (GMSL) indicator is calculated from the accurate and stable measurements of four reference altimeter missions, namely TOPEX/Poseidon (T/P), Jason-1, Jason-2 and Jason-3. All four reference missions are flying (or have flown) over the same historical ground track on a 10-day repeat cycle. They all, have been precisely inter-calibrated (Zawadzki and Ablain, 2016) to ensure the long term stability of the sea level measurements. Six research groups (AVISO/CNES, SL_cci/ESA, University of Colorado, CSIRO, NASA/GSFC, NOAA) process the sea level raw data provided by satellite altimeter to estimate the GMSL time series on a 10-day basis (Figure 1). The six different estimates of the GMSL record show small differences. The differences range between 1 and 2 mm at inter-annual time scales (1 to 5-year time scales) and between ±0.15 mm/yr in terms of trend over the period 1993-2017. The spread across these estimates is due to the use of different processing technique, different versions of ancillary data and different interpolation methods applied by the different groups (Henry et al., 2014; Masters et al., 2012). This spread is smaller than the real uncertainty in the sea level trend because all groups use similar methods and corrections to process the raw data and thus several sources of systematic uncertainty is not accounted for in the spread.

In a previous study (Ablain et al., 2009) proposed a realistic estimate of the uncertainty in the GMSL trend over 1993-2008 using an error budget approach. They identified the radiometer wet tropospheric correction as the main source of error. They also identified the orbit determination, the inter-calibration of altimeters and the estimate of the altimeter range, sigma-0 and significant wave height (mainly on TOPEX/Poseidon) as significant sources of error. When all terms were accounted for, they found that the uncertainty on the trend over 1993-2008 was ±0.6 mm/yr within a 90% confidence level. This is larger than the uncertainty of ±0.3 mm/yr over a 10-year period required by GCOS (GCOS, 2011). In the framework of the ESA Sea Level Climate Change Initiative (SL_cci ), significant improvements were made in the estimation of sea level from space (Ablain et al., 2015; Legeais et al., 2018; Quartly et al., 2017) to get closer to the GCOS requirements. New altimeter standards including new wet troposphere corrections, new orbit solutions, new atmospheric corrections and others were selected and applied to improve the sea level estimation. The GMSL trend uncertainties were then updated and estimated at different temporal and spatial scales (Ablain et al., 2015; Legeais et al., 2018). During the second altimetry decade, from 2002 to 2014, Ablain et al., (2015) estimated that the GMSL trend uncertainty was lower than ±0.5 mm/yr within a 90% Confidence Level (CL) for periods longer than 10 years.



In previous studies,  the uncertainty in GMSL have been estimated for long term trends (periods of 10 years or more, that start in 1993), interannual time scales (at time scales between 1 and 5 years) and annual time scales (Ablain et al., 2009, 2015),. This estimation of the uncertainty on three different time scales is a valuable first step but it is not sufficient, as it does not fully meet the needs of the scientific community. Indeed, for many climate studies there is a need for GMSL uncertainty over different time scales and over different time spans within the

25-year altimetry record. For instance, in sea level budget studies consisting in assessing the evolution of the different GMSL components, there is a need for GMSL uncertainty estimates at monthly time scales when we want to interpret GMSL monthly changes in terms of ocean mass changes (because ocean mass changes are resolved at monthly time scales since 2002 by the gravity recovery and climate experiment – GRACE – mission). In studies that estimate the Earth energy imbalance with the sea level budget approach, this is also the case (e.g.

Meyssignac et al., 2018). In detection attribution studies (e.g. Slangen et al., 2017), uncertainty in trend estimates are often needed but over different time spans than the ones addressed in  Ablain et al., 2015, 2009 and in Legeais et al., 2018. Sometimes it is the uncertainty on different metrics that is needed. For example recently (Dieng et al., 2017; Nerem et al., 2018) estimated the acceleration in GMSL over 1993-2017 and found a small acceleration of ~0.08 mm/yr² over the 25 year–long altimetry record. There is a need for the estimation of the

uncertainty in the GMSL acceleration to determine whether this acceleration is significant or not.

Note that here in this paper we focus on the uncertainty in the GMSL record that arises from the errors in the satellite altimetry instrument. This uncertainty can be used to quantify the measurement uncertainty on the GMSL record. This is an important piece of information for detection attribution studies that seek to estimate the GMSL

rise in response to anthropogenic activity. But this is not sufficient. In detection attribution studies the response of the GMSL to anthropogenic activity need to be separated from the response of the GMSL to the natural variability of the climate system because the latter is an additional source of uncertainty. Here we do not address this problem of separating the GMSL response to different sources of variability. We strictly focus on the instrument errors and the associated uncertainty.


The objective of this paper is to estimate the error variance-covariance matrix of the GMSL (on a 10-day basis) from satellite altimetry measurements. This error variance-covariance matrix provides a comprehensive description of the uncertainties in GMSL to users. It covers all time scales that are included in the 25-year long satellite altimetry record: from 10 days (the time resolution of the GMSL time series) to multidecadal time scales.

It also enables to estimate the uncertainty in any metrics derived from GMSL measurements such as trend, acceleration or other moments of higher order.



To estimate the error variance-covariance matrix, we use an error budget approach at global scale, on a 10-day basis, in which we consider all major sources of uncertainty in the altimetry measurement including the wet troposphere correction, the orbit solutions, the intercalibration of satellites and others. We also consider the temporal correlation between the different sources of uncertainty (section 2). Errors are characterized for each altimetry mission separately since different missions are affected by different sources of errors (section 2). On the basis of the error variance-covariance matrix we estimate the uncertainty in GMSL individual measurements on a 10-day basis (section 3) and the uncertainty in trend and acceleration over all periods included in the 25-year satellite altimetry record (1993-2017) (Section 4). Note that in this article all uncertainties associated to the GMSL are reported with a 90% CL unless stated otherwise.

## 1   GMSL data series

The six main groups that provide satellite altimetry based GMSL estimates (AVISO/CNES, SL_cci/ESA, University of Colorado, CSIRO, NASA/GSFC, NOAA) use 1-Hz altimetry measurements from T/P, Jason-1, Jason-2 and Jason-3 missions from 1993 to 2018 (1993-2015 for SL_cci/ESA). Each group process the 1-Hz data with geophysical corrections to correct the altimetry measurement for various aliasing, biases and drifts (caused for example by different atmospheric condition, different sea states, by ocean tides and others (see Ablain et al., 2009 for more details). Then they average spatially the data over each 10-day orbital cycle to provide GMSL estimates on a 10-day basis. Differences among GMSL estimates from different groups arise from different data editing, differences in the geophysical corrections, and differences in the method used to spatially average individual measurements during orbital cycles (Henry et al., 2014; Masters et al., 2012).

Recently, comparisons of the GMSL time series derived from satellite altimetry with independent estimates based on tide gauge records (Valladeau et al., 2012; Watson et al., 2015) or on the combination of the contribution to sea level from thermal expansion, land ice melt and land water storage (Dieng et al., 2017) showed that there was a drift in the GMSL record over the period 1993-1998. This drift is caused by an erroneous onboard calibration correction on TOPEX altimeter side-A (noted TOPEX-A). TOPEX-A was operated from launch in october 1992 to the end of January 1999.Then TOPEX side-B altimeter (noted TOPEX-B) took over in February 1999 (Beckley et al., 2017). The impact on the GMSL changes is -1.0 mm/yr between January 1993 and July 1995, and +3.0 mm/yr between August 1995 and February 1999, with an uncertainty of ±1.7 mm/yr (within a 90%CL, (Ablain, 2017)).

Without taking into account the TOPEX-A drift correction, the differences between all GMSL time series are small. The maximum trend difference between all time series over 1993-2017 is lower than 0.15 mm/yr, representing less than 5% of the GMSL trend. The differences observed at interannual time scales are also small



(<2 mm). By correcting the drift of TOPEX-A using either of the available empirical corrections (WCRP Global
       Sea Level Budget Group, 2018) the differences among solutions remain the same (the difference between
       empirical corrections being smaller than the difference between the raw GMSL time series).. Therefore, the
       choice of one or the other GMSL record is not decisive in this study, whose purpose is to characterize the
       uncertainties. Hereafter we use the GMSL AVISO record. The corresponding altimeter standard corrections and
the GMSL processing methods are described on the AVISO website (https://www.aviso.altimetry.fr/msl/).

## 2    Altimetry GMSL error budget

       This section describes the different errors that affect the altimetry GMSL record. It builds on the GMSL error
       budget presented in (Ablain et al., 2009) and extends this work by taking into account new altimeter missions
(Jason-2, Jason-3) and recent findings on altimetry error estimates. Three types of errors are considered: a)
       biases in GMSL between successive altimetry missions which are characterized by bias uncertainties ($\pm\Delta$) at a
       given time (t); b) drifts in GMSL characterized by a trend uncertainty ($\pm\delta$) and c) other measurement errors which
       exhibit time-correlation (so called residual time correlated errors here after). The residual time correlated errors
       are characterized by their standard deviation ($\sigma$) and correlation time scale ($\lambda$). All altimetry errors identified in
this study are summarized in Table 1 and detailed hereafter. Note that all uncertainties reported in Table 1 are
       gaussians and they are given at the 1-sigma level (i.e. we provide the standard deviation of the Gaussian, noted
       1-$\sigma$ hereafter

       Biases can arise between the GMSL record of two successive satellite missions like between T/P and Jason-1 in
May 2002, Jason-1 and Jason-2 in October 2008 and between Jason-2 and Jason-3 in October 2016. These
       biases are estimated during dedicated 9-month inter-calibration phases when a satellite altimeter and its
       successor fly over the same track, 1 minute apart. During the inter-calibration phases the bias is estimated and
       corrected for. Different missions show different biases but the uncertainty in the bias correction is the same for all
       inter-calibration phases and amounts: ±0.5 mm (Zawadzki and Ablain, 2016). The situation is different for the
switch from TOPEX-A to TOPEX-B in February 1999 because it was impossible to do any inter-calibration phase
       between the two sides of TOPEX (as both instruments were flying on the same spacecraft). For the switch, we
       assume that the uncertainty in GMSL is larger and is about 2 mm (Zawadzki and Ablain, 2016).

       Drifts may occur in the GMSL record because of drifts in TOPEX–A and TOPEX-B radar instruments, because of
drifts in the International Terrestrial Reference Frame (ITRF) realization in which altimeter orbits are determined
       or because of drifts in the Glacial Isostatic Adjustment (GIA) correction applied to the GMSL record. As explained



before the TOPEX-A record shows a spurious drift due to an erroneous onboard calibration correction of the altimeter (Beckley et al., 2017). This drift is corrected by different empirical approaches (Ablain, 2017; Beckley et al., 2017; Dieng et al., 2017) that are all affected by a significant uncertainty. With a comparison against an

independent GMSL estimate based on tide gauge records (Ablain, 2017), we estimate this uncertainty to be ±0.7 mm/yr (1-$\sigma$ level ) over the TOPEX-A period (1993-1998). For the TOPEX-B record, no GMSL drift has been reported, but (Ablain et al., 2012) showed significant Sigma-0 instabilities of the order of 0.1 dB, which generates through the sea-state bias correction an uncertainty of ±0.1 mm/yr (1$\sigma$ level)  in the GMSL record over the TOPEX-B period (February1999-April 2002). Concerning the ITRF realization, (Couhert et al., 2015) showed that

the uncertainty on the ITRF realization drift generates an uncertainty of ±0.1 mm/yr (1-$\sigma$ level) on the GMSL trend over 1993-2015. We adopt this value here for the whole period 1993-2017. For the uncertainty on the GIA correction applied to the GMSL, we use the value of 0.05 mm/yr (1-$\sigma$ level)  over the altimetry period from Spada (2017). Combining the uncertainty on the GMSL trend over 1993-2017 from GIA and ITRF and assuming that they are not correlated yield an uncertainty on the GMSL trend of ±0.12 mm/yr over 1993-2017 (1-$\sigma$ level).


Residual time correlated errors are separated into two different groups depending on their correlation time scales. The first group gathers errors with short correlation time scales i.e. lower than 2 months and between 2 months and 1 year. The second group gathers errors with long correlation time scales between 5 and 10 years. In the first group errors are mainly due to the geophysical corrections (e.g. ocean tides, atmospheric corrections…), the

altimeter corrections (e.g. sea-state bias correction, altimeter ionospheric corrections), the orbit calculation, and potential altimeter instabilities (e.g. altimeter range and sigma-0 instabilities). At time scales below 1 year, the variability of the corrections' time series is dominated by errors such that the variance of the error in each correction is estimated by the variance of the correction's time series. For errors with correlation time scales lower than 2 months, we estimate the standard deviation ($\sigma$) of the error from the correction's time series filtered

with a 2-month high-pass filter. As the standard deviation of the errors depends on the different altimeter missions, the standard deviation has been estimated separately for each altimeter mission. We find $\sigma$ = 1.7 mm over the T/P period, $\sigma$ = 1.5 mm over the Jason-1 period, and $\sigma$ = 1.2 mm over the Jason-2/3 period. For errors with correlation time scale between 2 months and 1 year, we used the same approach and filtered the correction time series with a pass-band filter. In this case we find $\sigma$ = 1.3 mm over the T/P period, $\sigma$ = 1.2 mm over the

Jason-1 period, and $\sigma$ = 1.0 mm over the Jason-2/3 period. Not surprisingly, the highest errors are obtained for T/P, and the lowest ones for Jason-2/3. This is because of 1) larger altimeter range instabilities in T/P (Ablain et al., 2012; Beckley et al., 2017),  2) the presence of a 59-day signal error in the altimeter range of T/P (Zawadzki et al., 2018), and 3) because of the deterioration in the performance of atmospheric corrections in the early years of the altimetry era (Legeais et al., 2014). Note that Jason-1 shows also higher errors than Jason-2 and Jason-3

at time scales below 1 year (Couhert et al., 2015).



In the second group of residual time correlated errors, errors are due the on-board microwave radiometer calibration that yield instabilities in the wet troposphere correction, and also to the orbit calculation (Couhert et al., 2015). Because these errors are correlated at time scale longer than 5 years they can not be estimated with the standard deviation of the correction time series, the correction time series being too short (25-year long) to
sample the time correlation. For this group of residual time correlated errors we use simple models to represent the time correlation of the errors. For the wet troposphere correction, several studies (Fernandes et al., 2015; Legeais et al., 2014; Thao et al., 2014) have identified long-term differences among the corrections computed from the different microwave radiometers and from different atmospheric reanalyses (e.g. ERA-interim reanalyzes (Dee et al., 2011). These studies report a difference in the wet tropospheric correction for GMSL in
the range of ±0.2-0.3 mm/yr for periods of 5 to 10 years. Here we adopt a conservative approach and we model the error in wet tropospheric correction with a correlated error at 5 years with a standard deviation of 1.2 mm (1 $\sigma$ level). The correlation is modeled with a gaussian attenuation based on the wavelength of the errors: $e^{\frac{-1}{2}\left(\frac{t}{\lambda}\right)^2}$ with $\lambda$= 5 years. In terms of trends, this residual time correlated error generates an uncertainty of ±0.2 mm/yr over 5-year periods. For the error in the orbit calculation, comparisons of different orbit solutions showed differences of
±0.05 mm/yr on 10 year time scales due to errors in the modelling of the Earth time varying gravity field (Couhert et al., 2015). We model this error with a correlated error at 10-year time scale with a standard deviation of 0.5 mm (1-$\sigma$ level). The correlation is modeled by the same gaussian distribution as before with $\lambda$=10 years. In terms of trends, it corresponds to an uncertainty of ±0.05 mm/yr over 10-year periods.

In the next section these different terms of the GMSL error budget are combined together to build the error
variance-covariance matrix. Note that the different terms of the altimeter GMSL error budget described here are based on the current knowledge of altimetry measurement errors. As the altimetry record increases in length with new altimeter missions, the knowledge of the altimetry measurement also increases, and the description of the errors improves. This implies that the error variance-covariance matrix is expected to improve and change in the future.

**3   The GMSL error variance-covariance matrix**

In this section we derive the error variance-covariance matrix ($\Sigma$) of the GMSL from the error budget described in section 2. We assume that all error sources presented in Table 1 are independent from each other. Thus the $\Sigma$ matrix is the sum of the individual variance-covariance matrix of each error source $\Sigma_i$ in the error budget (see Figure 2). Each $\Sigma_i$ matrix is calculated from a large number of random draws (> 1000) of simulated error signal
using the model described in section 2 (either a bias, drift or time correlated signal) fed with a standard normal distribution.



The resulting shape of each individual $\Sigma_i$ matrix depends on the type of error (bias, drift or time correlated signal, see Figure 2). For biases, the $\Sigma_i$ matrix takes the shape of constant square blocks each side of the time occurrence of the bias correction (see for example the square matrix for TOPEX-A and TOPEX-B on the low left corner of Figure 2 along the diagonal). This shape in square block means that the error in the bias correction generates an error on the GMSL which is fully correlated along time before and after the bias correction time, but which is not correlated along time for dates that are apart of the bias correction time. This is consistent with what we expect from a bias correction error. Note that in this article (and in climate change studies in general) we are interested only in GMSL changes, trends or acceleration but not on the time mean GMSL (which is the absolute reference of GMSL). Thus, we have removed from the GMSL time series the temporal mean over 1993-2017. The reference of the GMSL is thus arbitrary and assumed to be perfectly known. This is the reason why the reference of the GMSL is not affected by the biases correction error here.

For drifts, the $\Sigma_i$ matrix take the shape of a horse saddle. This is because an error on the GMSL drift over a given period generates errors on the GMSL time series which are correlated when they are close in time and anti-correlated when they are on opposite side of the drift period.

For residual time correlated errors, the $\Sigma_i$ matrix take the shape of a diagonal matrix with off diagonal terms of smaller amplitude. The more the off-diagonal terms are far from the diagonal the more they are attenuated. The attenuation rate is a Gaussian attenuation based on the wavelength of the time correlated errors ($e^{\frac{-1}{2}\left(\frac{t}{\lambda}\right)^2}$), with various time-scales λ.

All individual $\Sigma_i$ matrix are summed up together to build the total error variance-covariance matrix $\Sigma$ of the altimetry-derived continuous GMSL record over 1993-2018 (see Figure 2). As expected, the dominant terms of the matrix are on the diagonal. They are largely due to the different sources of errors with correlation time scales below 1 year (first group of errors in section 2). The diagonal terms are the highest at the beginning of the altimetry period when T/P was at work. This is because of larger altimeter range instabilities in T/P, the presence of a 59-day signal error on the altimeter range of T/P and poorer performance of atmospheric corrections in the early years of the altimetry era (Legeais et al., 2014). The dominant off-diagonal terms are also found during the T/P period (in the lower left corner of the matrix, see Figure 2). The terms are induced by the TOPEX-A trend error and the large bias correction uncertainty between TOPEX-A and TOPEX-B (because of the absence of inter-calibration phase between TOPEX-A and TOPEX-B).

## 4    GMSL uncertainty envelope



We estimate the GMSL uncertainty envelope from the square root of the diagonal terms of $\Sigma$ (see Figure 3). As expected, the GMSL time series shows a larger uncertainty during the T/P period (5 mm to 8 mm) than during the
Jason period (close to 4 mm). The bias correction uncertainty between TOPEX-A and TOPEX-B in February 1999 is also clearly visible with a 1-mm drop in the uncertainty after the switch to TOPEX side-B. Note that the uncertainty envelope has a parabolic shape and shows smaller uncertainties during the beginning of the Jason-2 period (3.5 mm around 2008) than over the Jason-3 period (4.5 mm).  This is not because Jason-1 and Jason-2 errors are smaller than Jason-3's errors. Actually Jason 2 and Jason-3 errors are slightly smaller than Jason-1
errors thanks to better orbit determination.  The uncertainties are smaller during the Jason-1 and Jason-2 period because this period is in the center of the record. It benefits from prior and posterior data that covariate and help in reducing the uncertainty when they are combined together. In contrast the Jason-3 period is located at the end of the record and does not benefits from posterior data to help reduce the uncertainty.

On Figure 4 we superimpose the GMSL time series (average of the GMSL time series in Figure 1) and the associated uncertainty envelop. For the TOPEX-A period we test 3 different curves with three different corrections based on the removal of the Cal-1 mode (Beckley et al., 2017), the comparison with tide gauges (Ablain, 2017; Watson et al., 2015), or based on a sea level closure budget approach (Dieng et al., 2017).  The uncertainty envelop is centered on the record corrected  for TOPEX-A drift with the correction based on (Ablain et
al., 2017). As expected, all empirically corrected GMSL records are within the uncertainty envelop.

## 5    Uncertainty in GMSL trend and acceleration

The variance-covariance matrix can be used to derive the uncertainty on any metric based on the GMSL time
series. In this section we use the error variance-covariance matrix to estimate the uncertainty on the GMSL trend and the GMSL acceleration over any period of 5 years and more within 1993-2017.
Recently, several studies (Dieng et al., 2017; Nerem et al., 2018; Watson et al., 2015; WCRP Global Sea Level Budget Group, 2018) found a significant acceleration in the GMSL record from satellite altimetry  (after correction of the TOPEX-A drift) . The presence of an acceleration in the record should not change the estimation of the
trend when estimated with a least square approach. However it can affect the estimation of the uncertainty on the trend. To cope with this issue we address here at the same time the estimation of the trend and the estimation of the acceleration in the GMSL record. To this objective we use a second order polynomial as predictor. Considering the GMSL record has n observations, let X be an n × 3 predictor where the first column contains only ones (representing the constant term), the second column contains the time vector (representing the linear term)



and the third column contains the square of the time vector (representing the squared term). Let y be an n × 1 vector of independent observations of the GMSL. Let $\epsilon$ be an n × 1 vector of disturbances (GMSL non-linear and non-quadratic signals) and errors. Let β be the 3 × 1 vector of unknown parameters that we want to estimate, namely the GMSL y-intercept, the GMSL trend and the GMSL acceleration. Our linear regression model for the estimation of the GMSL trend and acceleration will thus be

$$y = X\beta + \epsilon.$$

with

$$\epsilon \sim N(0, \Sigma)$$

where $\Sigma$ is the variance-covariance matrix of the observation errors (estimated in the previous section). $\Sigma$ is different from the identity because of the correlated noise (see section 2).

The most common method to estimate the GMSL trend and acceleration is the Ordinary Least Squares (OLS) estimator in its classical form (Cazenave and Llovel, 2010; Dieng et al., 2015; Masters et al., 2012; Nerem et al., 2018). This is also the most common method to estimate trends and accelerations in other climate essential variables (Hartmann, et al., 2014 and references therein). For this reason we turn here to the OLS to fit the linear regression model. The estimator of β with the OLS approach, noted $\hat{\beta}$ is

$$\hat{\beta} = (X^t X)^{-1} X^t y.$$

In most cases, $\epsilon$ follows a N(0,σ² I) distribution, which implies that $\hat{\beta}$ follows a Normal Law

$$\hat{\beta} = N(\beta, \sigma^2 (X^t X)^{-1})$$

The issue with this common framework is that the uncertainty of the trend and acceleration estimates does not
take into account the correlated errors of the GMSL observations.

To address this issue, we use a more general formalism to integrate the GMSL error in the trend uncertainty estimation, following (Ablain et al., 2009; Ribes et al., 2016), and IPCC AR5 (Hartmann, et al., 2014, see in particular Box 2.2 and Supplementary Material). The OLS estimator is let unchanged (and is still unbiased), but its distribution is revised to account for Σ, leading to:

$$\hat{\beta} = N(\beta, (X^t X)^{-1} (X^t \Sigma X)(X^t X)^{-1})$$

Note that this estimate is known to be less accurate than the General Least Square estimate (GLS, which is the optimal estimator in the case where Σ ≠ I) in terms of the mean square error, because its variance is larger. A generalized least square estimate would probably help in narrowing slightly the trend uncertainty but the
difference would likely be small as the GMSL time series is almost linear in time. Important advantages of using OLS here are (i) OLS is consistent with previous estimators of GMSL trends as well as estimators of trends in



other essential climate variable than GMSL (e.g. Hartmann, et al., 2014), and (ii) the OLS best-estimate does not depend on the estimated variance-covariance matrix Σ.

Based on the matrix Σ defined in the previous section, and the OLS solution proposed before, we now estimate
the GMSL trend (mm/yr) and acceleration (mm/yr²) uncertainties for any time span included in the period 1993-2017. Results are synthetically displayed in

Figure 5 for trends and in Figure 6 for accelerations. On Figure 5, the top of the triangle indicates that the GMSL trend uncertainty over 1993-2017 is ±0.4 mm/yr (CL 90%) and that the GMSL acceleration uncertainty over the
same period is ±0.07 mm/yr² (CL 90%, Figure 6). The GMSL acceleration uncertainty estimate is consistent with results of Watson et al., 2015, on the January 1993 to June 2014 time period where they find an uncertainty of $\pm 0.058\, mm.yr^{-2}$ at $1\sigma$ which corresponds to ±0.096 mm/yr² at the 90% confidence level. This is slightly larger than  Nerem et al. (2018) estimate which is ±0.025 mm/yr² at  1-σ level on the full 25-year altimetry era which corresponds to ±0.041 mm/yr² at 90% confidence level. But  Nerem et al. (2018) estimate is likely underestimated
as they only consider omission errors. The GMSL acceleration uncertainties have been calculated for all periods of 10 years and more within 1993-2017 (Figure 6). As expected, uncertainties tend to increase when the period length decreases. At 10 years, the GMSL acceleration uncertainties are ranging from ±0.3 mm/yr² over the T/P period to ±0.25 mm/yr² over the Jason period. At 20 years they range between ±0.12 and ±0.08 mm/yr².

A cross-sectional analysis of the 10-year horizontal line on Figure 5 shows that the GMSL trend uncertainties over 10 years periods decreased from 1.0 mm/yr over the first decade to 0.5 mm/yr over the last one. The larger uncertainty over the first decade is mainly due to the TOPEX-A drift error but also to the large intermission bias uncertainty between TOPEX-A and TOPEX-B and to a lesser extent to the improvement of GMSL accuracy with Jason-2 and Jason-3. Note that the current GCOS requirement of 0.3 mm/yr uncertainty over 10 years (GCOS,
2011) is not met at the 90% confidence level. But the recent record over the last decade based on the Jason series is close to meet the GCOS requirement with a 90% CL. Figure 5 can also be analysed by following the sides of the triangle. The results of this analysis are plotted on Figure 7. The plain line corresponds to the left side, read from bottom left to the top of the triangle. The dashed line corresponds to the right side, read from bottom right to the top of the triangle. As expected, both curves show a reduction of the trend uncertainty as the
period over which trends are computed increases from 2 to 25 years. The difference between the two lines shows the reduction of GMSL errors thanks to the improvement of the measurement in latest altimetry missions. The lowest trend uncertainty is obtained with the last 20 years of the GMSL record: 0.35 mm/yr.

Figure 8 indicates the periods for which the acceleration in sea level is significant at the 90% confidence level.
The acceleration is visible at the end of the record for periods of 10 years and longer. The GMSL acceleration is



0.12 mm/yr² with an uncertainty of 0.07 mm/yr² at 90% confidence level over the 25-year altimetry era. This proves that the acceleration observed in the GMSL evolution is statistically significant. It is worth noting that the different empirical TOPEX-A corrections yield very similar results (0.126 mm/yr² (Ablain, 2017) ; 0.120 mm/yr² (Dieng et al., 2017; Watson et al., 2015), 0.114 mm/yr² (Beckley et al., 2017). This acceleration at the end of the

record is due to an acceleration in the contribution to sea level from Greenland and from other contributions but to a lesser extent (Chen et al., 2017; Dieng et al., 2017; Nerem et al., 2018). A small acceleration is also visible during the period 1993-2005 at the beginning of the record. This acceleration is likely due to the recovery from the Mount Pinatubo eruption in 1991 (Fasullo et al., 2016).

Figure 9 indicates the period for which the trend in sea level is significant at the 90% confidence level. In periods
where the acceleration is not significant the second order polynomial that we used as predictor to estimate the trend and the acceleration does not hold anymore in principle. For these periods we should turn to a first order polynomial. The use of a first order polynomial does not affect the trend estimates. It only affects the trend uncertainty estimates. We checked for differences in trend uncertainty when using either second order or first order polynomial predictors. We found that the differences are negligible (not shown). Figure 9 indicates that for
periods of 5 years and longer the trend in GMSL is always significant at 90% CL over the whole record. At the end of the record the trend tend to increase which is consistent with the acceleration plot on Figure 6. Over the 25 years of satellite altimetry we find a sea level rise of .3.35 ± 0.4 mm/yr (90% CL), after correcting for the TOPEX-A GMSL drift. The differences due to the different TOPEX A corrections is negligible (<0.05 mm.yr$^{-1}$).

## 6   Conclusions

In this study we have estimated the full GMSL error variance-covariance matrix over the satellite altimetry period. The matrix is available online (see section data). It provides to users a comprehensive description of the GMSL errors over the altimetry period.  This matrix is based on the current knowledge of altimetry measurement errors. As the altimetry record increases in length with new altimeter missions, the knowledge of the altimetry measurement also increases, and the description of the errors improves. Consequently the error variance-
covariance matrix is expected to change and improve in the future – hopefully with a reduction of measurement uncertainty in new products.

The uncertainty of the GMSL computed here shows the reliability of altimetry measurements to accurately describe the evolution of the GMSL on all time scales from 10 days to 25 years. It also shows the reliability of
altimetry measurements to estimate sea level trends and now accelerations. Along the altimetry record, we find that the uncertainty in each individual GMSL measurement decreases with time. It is smaller during the Jason era (2002-2018) than during the T/P period (1993-2002). Over the entire altimetry record, 1993-2017, we estimate





the GMSL trend to 3.35 ± 0.4 mm/yr (90% CL, after correcting the TOPEX-A GMSL drift). We detect also a significant GMSL acceleration over the 25-year period at 0.12 ±0.07 mm/yr² (90% CL).


Several assumptions have been made in this study that could be improved in the future. First, the modelling of altimeter errors should be regularly revisited and improved to take into account a better knowledge of errors (e.g. stability of wet troposphere corrections) and to consider future altimeter missions (e.g. Sentinel-3 and Sentinel-6 missions). With regards to the mathematical formalism, OLS method has been applied because it is the most

common approach used in the climate community to calculate trends in any climate data records. However this is not the optimal linear estimator. The use of a Generalized Least Square approach should involve some narrowing of trend or acceleration uncertainty. Another topic of concerns is the consideration of the internal and forced variability of the GMSL. Here we only considered the uncertainty in the GMSL due to the satellite altimeter instrument. In a future study it would be interesting to consider the partition of the GMSL into the forced response

to anthropogenic forcing and the natural response to natural forcing and to the internal variability. Estimating the natural GMSL variability (e.g. using models) and considering it as an additional residual time correlated error, would allow to calculate the GMSL trend and acceleration representing the long-term evolution of GMSL in relationship with anthropogenic climate change.

**Data**

The global mean sea level error variance-covariance matrix is available online at **https://doi.org/10.17882/58344** (Ablain et al., 2018).

**Acknowledgment**

This work was carried out as part of the Sea Level CCI (SL_cci) project (Climate Change Initiative program) supported by ESA and the SALP (Service d'Altimétrie et de Localisation Précise) project supported by CNES for

several years. We would also like to thank all contributors to these two projects, with special recognition to Jérôme Benveniste, technical officer of the SL_cci project at ESA, and Thierry Guinle, SALP project manager at CNES.

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



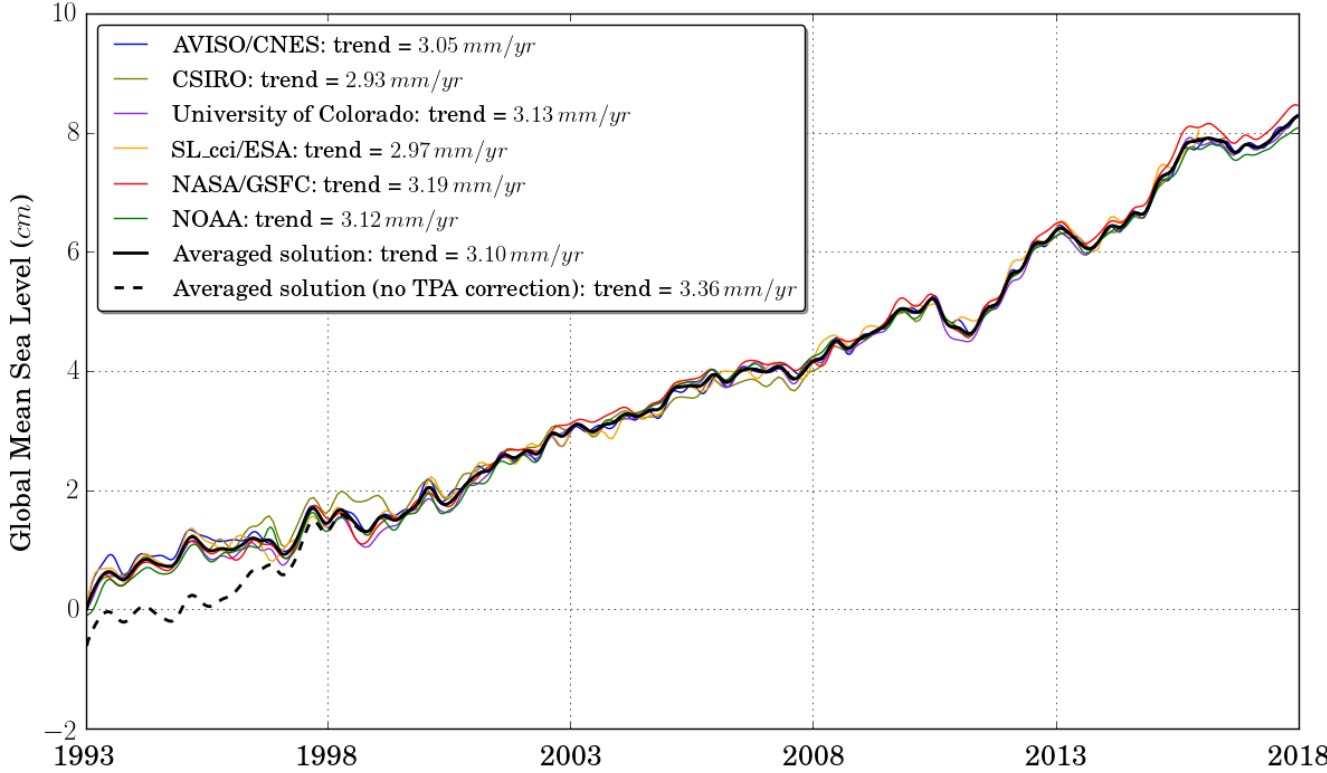

Figure 1: Evolution of GMSL time series (corrected for TOPEX-A drift using (Ablain, 2017) TOPEX-A correction)
from 6 different groups (AVISO/CNES, CSIRO, University of Colorado, SL_cci/ESA, NASA/GSFC, NOAA)
products. The SL_cci/ESA covers period January 1993 to December 2016 while all other products cover the full
25-year period (January 1993 to December 2018). Seasonal (annual and semi-annual) signals have been
removed and a 6-month smoothing is applied.. An averaged solution is computed from the 6 groups. GMSL time
series have the same average on the 1993-2015 period (common period) and the averaged solution starts at 0 in
1993. The average solution without TOPEX-A correction is also represented. A GIA correction of -0.3 mm/yr has
been subtracted to each data set.





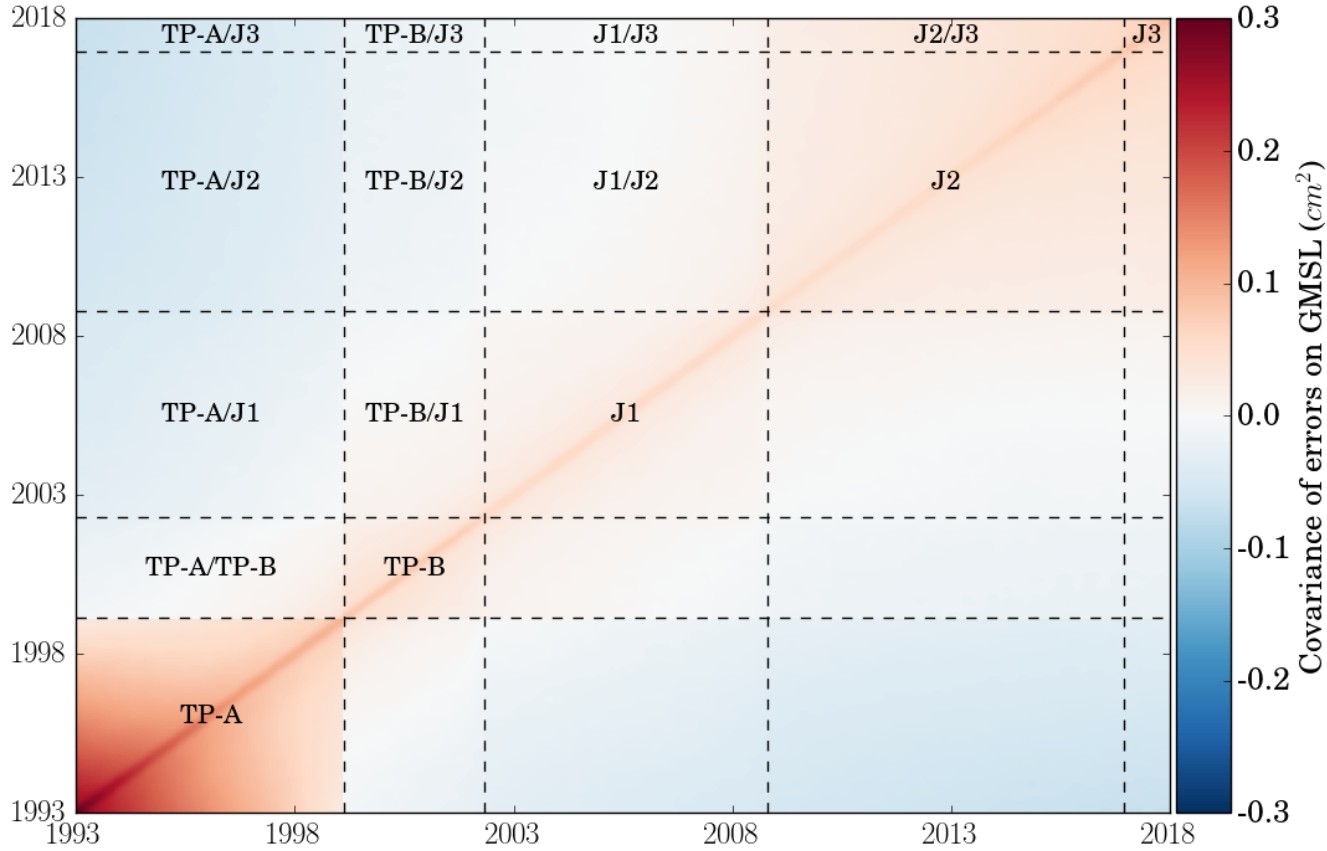

Figure 2: Error variance-covariance matrix of altimeter GMSL on the 25-years period (January 1993 to December 2017).



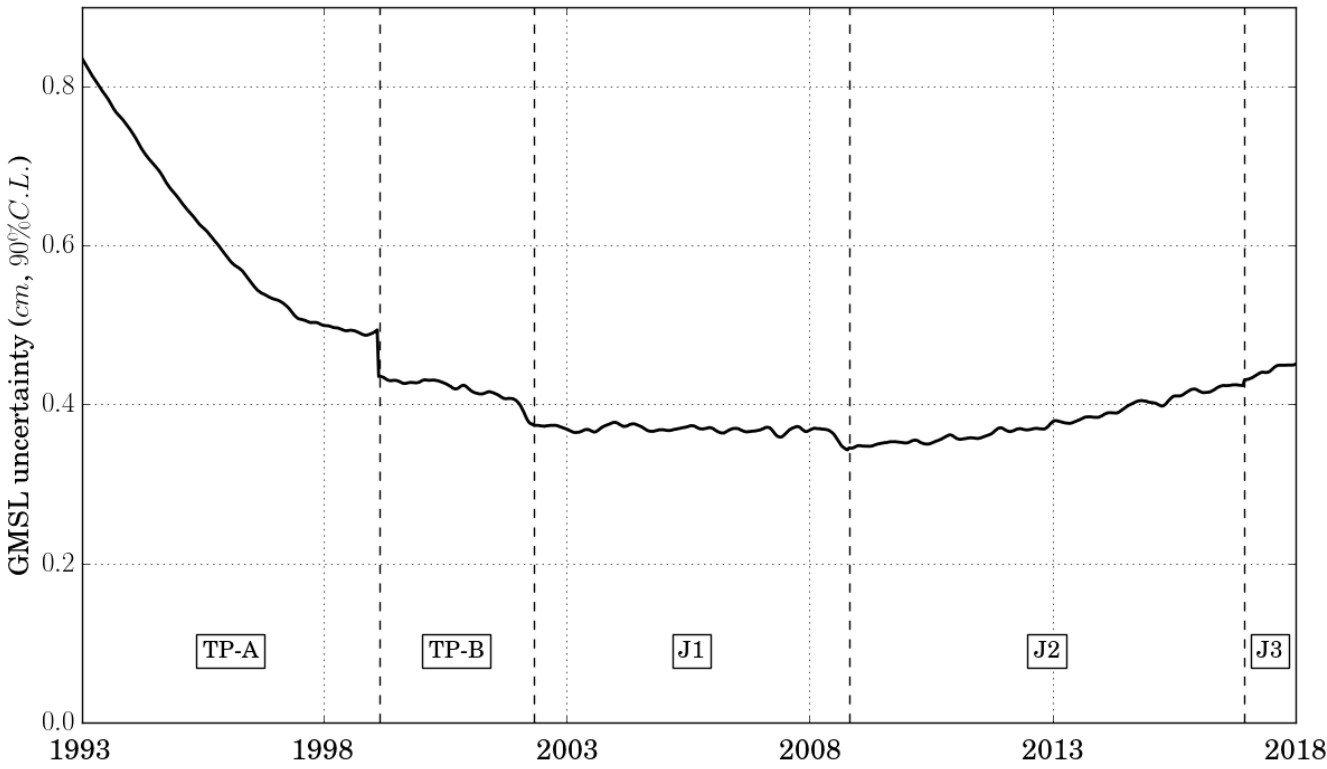

Figure 3: Evolution on time of GMSL measurement uncertainty within a 90 % confidence level (i.e. 1.65$\sigma$) on the 25-years period (January 1993 to December 2017).




Figure 4: Evolution of the AVISO GMSL with different TOPEX-A corrections. On the black, red and green curves, the TOPEX-A drift correction is applied respectively based on (Ablain, 2017), (Watson et al., 2015), (Dieng et al., 2017) and (Beckley et al., 2017). The uncertainty envelope, as well as trend and acceleration uncertainties are given at 90% confidence level (i.e. 1.65σ). Seasonal (annual and semi-annual) signals removed and 6-month smoothing applied; GIA correction also applied.






Figure 5: GMSL trend uncertainties (mm/yr) estimated for all altimeter periods within the 25-years period (January 1993 to December 2017). The confidence level is 90 % (i.e. $1.65\sigma$). Each colored pixel represents respectively the half-size of the 90% confidence range in GMSL trend. Values are given in mm/yr. The vertical axis indicates the length of the period (ranging from 1 to 25 years) considered in the computation of the trend while the horizontal axis indicates the center date of the period (for example 2000 for the 20-year period 1990-2009).





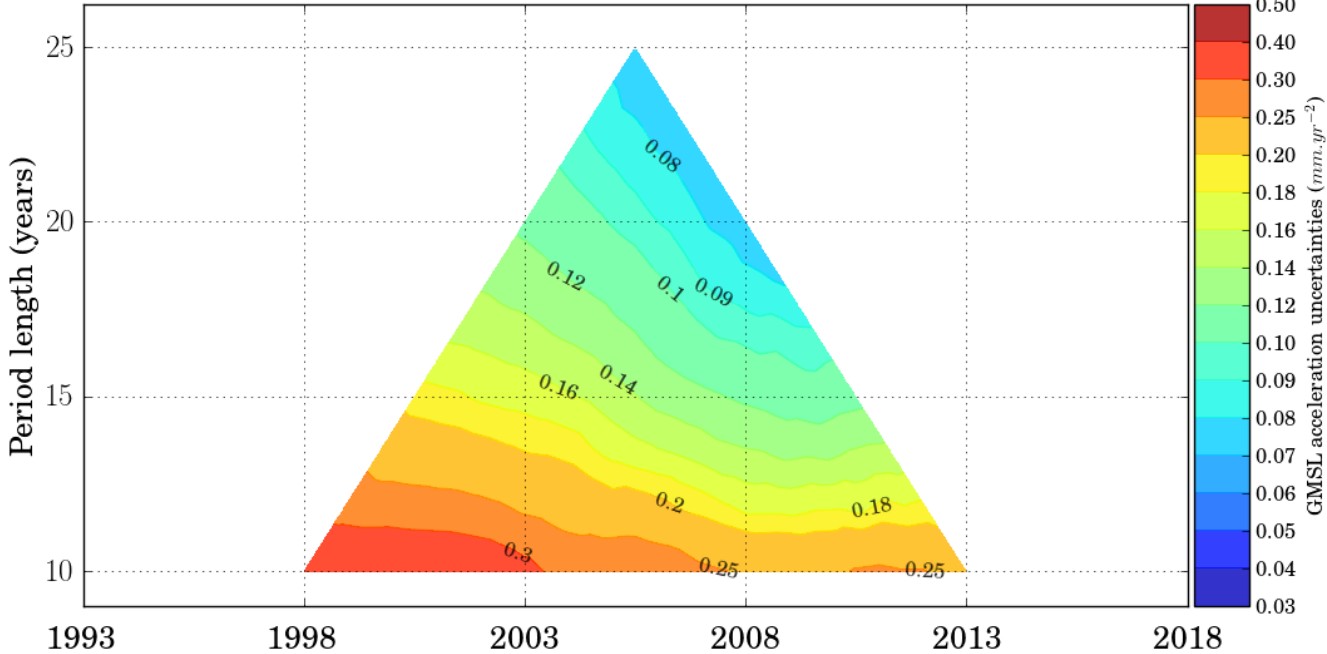


Figure 6: GMSL acceleration uncertainties (mm/yr²) estimated for all the altimeter periods within the 25-years period (January 1993 to December 2017). The confidence level is 90 % (i.e. $1.65\sigma$). Each colored pixel represents respectively the half-size of a 90% confidence range in GMSL acceleration. Values are given in 570 mm/yr2. The vertical axis indicates the length of the period (ranging from 1 to 25 years) considered in the computation of the acceleration while the horizontal axis indicates the center date of the period (for example 2000 for the 20-year period 1990-2009).




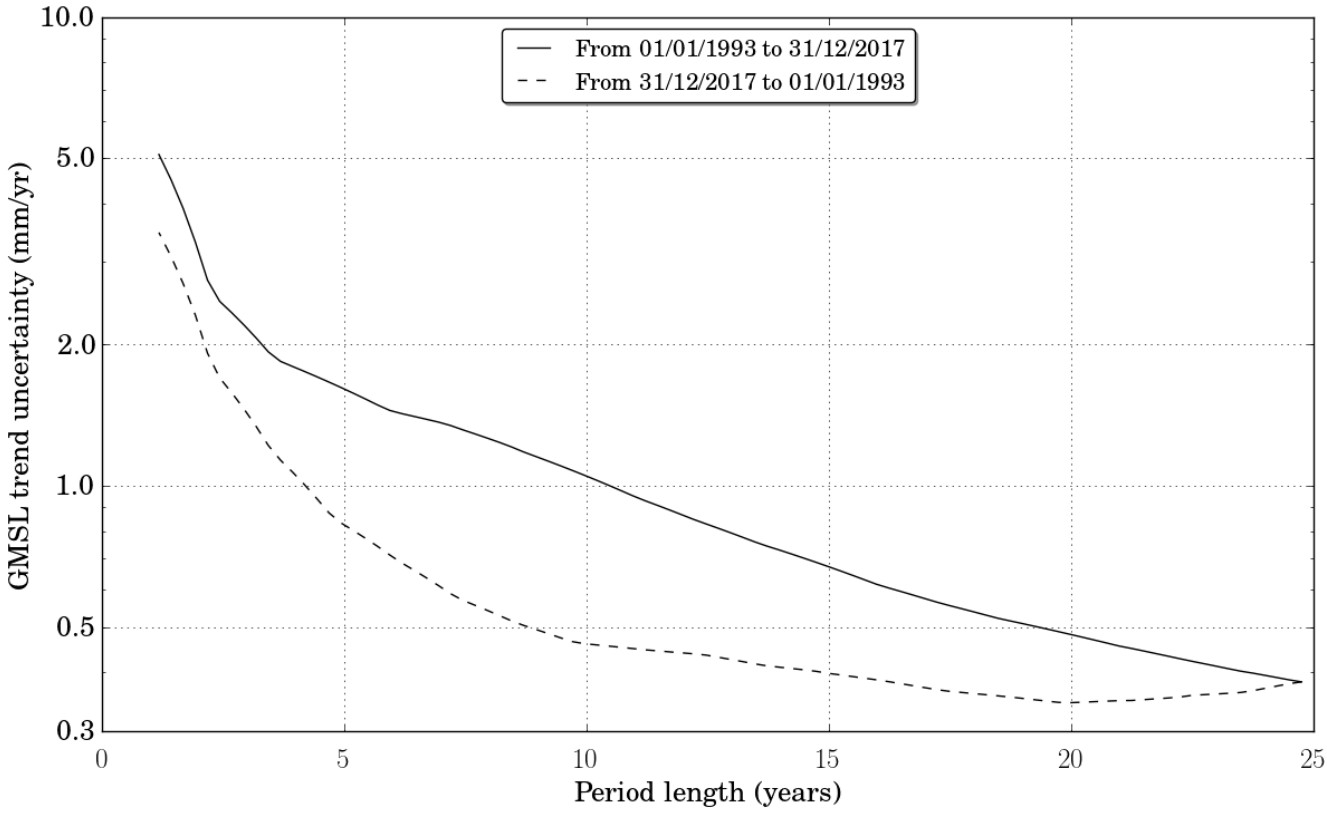


Figure 7: Evolution of GMSL trend uncertainties (within a 90% confidence level, i.e. $1.65\sigma$) versus the altimeter period length from January 1993 to December 2017 on plain curve and from December 2017 to January 1993 on the dashed curve.




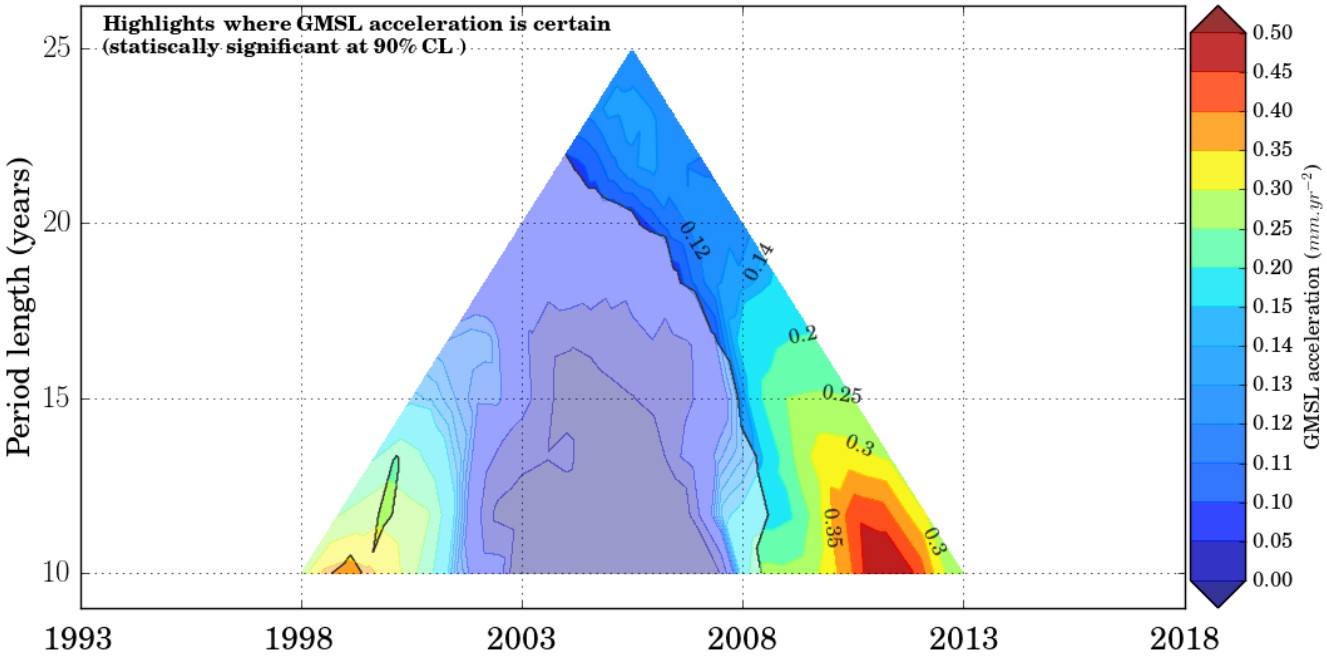

Figure 8: GMSL acceleration using the AVISO GMSL timeseries corrected for the TOPEX-A drift using the correction proposed by (Ablain, 2017): acceleration in shaded areas is not significant (i.e. lower than acceleration uncertainties at 90% confidence level). The length of the window (in years) is represented on the vertical axis and the central date of the window used (in years) is represented on the horizontal axis.








Figure 9: GMSL trends using the AVISO GMSL timeseries corrected for the TOPEX-A drift using the correction proposed by (Ablain, 2017).The length of the window (in years) is represented on the vertical axis and the central date of the window used (in years) is represented on the horizontal axis.




| Source of errors | Error category | Uncertainty level (at 1 $\sigma$) | References |
|---|---|---|---|
| High frequency errors: altimeter noise, geophysical corrections, orbits ... | Correlated errors (λ = 2 months) | $\sigma$ = 1.7 mm for TOPEX period<br>$\sigma$ = 1.5 mm for Jason-1 period.<br>$\sigma$ = 1.2 mm for Jason-2/3 period. | Calculation explained in this paper |
| Medium frequency errors: geophysical corrections, orbits .. | Correlated errors (λ = 1 year) | $\sigma$ = 1.3 mm for TOPEX period<br>$\sigma$ = 1.2 mm for Jason-1 period.<br>$\sigma$ = 1 mm for Jason-2/3 period. | Calculation explained in this paper |
| Large frequency errors: wet troposphere correction | Correlated errors (λ = 5 years) | $\sigma$ = 1.1 mm over all the period (⇔ to 0.2 mm/yr for 5 years) | (Legeais et al., 2014; Thao et al., 2014) |
| Large frequency errors: orbits (Gravity fields) | Correlated errors (λ = 10 years) | $\sigma$ = 1.12 mm over TOPEX period (no GRACE data)<br>$\sigma$ = 0.5 mm over Jason period (⇔ to 0.05 mm/yr for 10 years) | (Couhert et al., 2015; Rudenko et al., 2017) |
| Altimeter instabilities on TOPEX-A and TOPEX-B | Drift error | $\delta$ = 0.7 mm/yr on TOPEX-A period<br>$\delta$ = 0.1 mm/yr on TOPEX-B period | (Ablain, 2017; Beckley et al., 2017; Watson et al., 2015) |
| Long-term drift errors: orbit (ITRF) and GIA | Drift error | $\delta$ = 0.12 mm/yr over 1993-2017 | (Couhert et al., 2015; Spada, 2017) |
| GMSL bias errors to link altimetry missions together | Bias errors | $\Delta$ = 2 mm for TP-A/TP-B<br>$\Delta$ = 0.5 mm for TP-B/J1, J1/J2, J2/J3. | (Zawadzki et al., 2018) |

Table 1: Altimetry GMSL error budget given at 1-sigma