# Peer review of "Uncertainty in Satellite estimate of Global Mean Sea Level changes, trend and acceleration"

_Earth System Science Data, 2019_

## Referee Comment (RC1) · Anonymous Referee #1 · 24 May 2019

This paper provides the first estimate of an error variance-covariance matrix for altimeter measurements of global mean sea level rise. The authors then derive a 90% confidence interval of GMSL on a 10-day basis and estimate the trend and acceleration of GMSL over 5 year or longer intervals.

Overall the paper is easy to understand and could potentially provide a useful quantification of uncertainty. However, my primary concern is with the treatment of GIA uncertainty and the authors must address this.

The authors note that they use the Spada 2017 estimate of 0.05 mm/year for GIA uncertainty. This uncertainty estimate is for the GIA component due to the ongoing changes in the Earth's crust since the last glacial maximum (LGM) but does not include modern day melt contributions to GIA. As the authors are aware, the LGM-GIA

response is typically accounted for in altimeter-based estimates of GMSL by adding 0.3 mm/yr to the altimeter-derived estimate of GMSL. However, this estimate does not account for deformations of the ocean bottom due to modern melt, which can introduce biases in both the mean trend and acceleration term. See, for example, Frederikse et al. 2017 and Lickley et al. 2018. This correction need not be included if the authors wish to use altimeter measurements to estimate changes in sea surface height instead of sea level. However, the authors explicitly reference estimates of changes in sea level (lines 117- 120) where they compare altimeter estimates of GMSL to changes in ocean volume as measured by tide gauges, or the sum of the contributions to changes in ocean volume. To be consistent, I believe this additional source of GIA uncertainty should be accounted for. Alternatively, they could remove the GIA estimate altogether and state upfront that this is an estimate of the uncertainty in sea surface height and cannot be compared to volumetric changes in sea level.

Frederikse, T., Riva, R. E., & King, M. A. (2017). Ocean bottom deformation due to presentday mass redistribution and its impact on sea level observations. Geophysical Research Letters, 44(24).

Lickley, M. J., Hay, C. C., Tamisiea, M. E., & Mitrovica, J. X. (2018). Bias in estimates of global mean sea level change inferred from satellite altimetry. Journal of Climate, 31(13), 5263-5271.

Specific Comments:

There are a number of grammatical errors and issues with vocabulary choice throughout. Please check!

Here are a few examples: Line 41: add an s to "altimeter" Replace "confidence envelope" with "confidence interval" throughout Line 86: replace "the GMSL" with "GMSL". Line 96: Add "us" after "enables" and remove the "s" on "metrics"

Other issues: Line 307, should be '∼' not '=' Please label axes on Figure 5 and 9.

---

## Referee Comment (RC2) · Anonymous Referee #2 · 12 Jun 2019

**General comments**

In this paper the uncertainty in the satellite estimate of Global Mean Sea Level changes, particularly referring to the trend and the acceleration has been evaluated. I have read it with attention, finding that its quality is quite good, in my opinion. The English form is generally good but at some sections it needs to be further improved. Moreover, the research group appears to be qualified in the field of satellite oceanography. Nonetheless this, a moderate revision is still necessary for a further improvement of the paper's quality (see specific comments). The topic of Global Mean Sea Level and its relationships with climate changes has been deeply studied in marine geophysics and satellite oceanography (Ablain et al., 2015; 2017; Abraham et al., 2013; Allan et al., 2014; Aucan et al., 2017; Baki Iz et al., 2018; Beckley et al., 2010; 2017; Boen-

ing et al., 2012; Cazenave et al., 2014; Chambers et al., 2010; Chen et al., 2017a; 2017b; Church and White, 2006; 2011; Church et al., 2013; Clark and Primus, 1987; Conrad and Hager, 1997; Curry, 2018; Dahlen, 1976; Dangendorf et al., 2017; Davis and Mitrovica, 1996; Desai et al., 2015; Desbruyeres et al., 2016; Dieng et al., 2017; Esselborn et al., 2018; Farrell and Clark, 1976; Fasullo et al., 2013; 2016; Frederikse et al., 2017a; 2017b; 2018; Gardner et al., 2013; Gornitz et al., 2019; Gregory et al., 2013; Haigh et al., 2014; Hamlington and Thompson, 2015; Hamlington et al., 2013; 2016; 2017; Handoko and Hariyadi, 2018; Hay et al., 2015; Herring et al., 2019; Kay et al., 2014; Kendall et al., 2005; Kidwell et al., 2017; Lickley et al., 2018; Melachroinos et al., 2013; Merrifield et al., 2009; Milne and Mitrovica, 1996; Mitchum, 2000; Mitrovica and Milne, 2003; Mitrovica et al., 2001; Nerem and Fasullo, 2018; Nerem et al., 1999; 2010; 2018; Prandi et al., 2009; Ray et al., 2013; Shepherd et al., 2018; Slangen et al., 2016; 2017; Swart et al., 2015; Spada, 2017; Spada and Galassi, 2016; Tamisiea, 2011; Thompson et al., 2016; Trenberth et al., 2016; Vaughan et al., 2013; Wahr et al., 2015; Wang et al., 2017; Watkins et al., 2015; Watson et al., 2015; Wiese et al., 2016; Wouters et al., 2013). Due to the exceptional abundance of recent scientific literature addressing this research topic, I suggest perhaps to the authors to expand the discussion of their results, taking into account some of the scientific papers listed in the attached references, which have not considered in detail. This could be a general issue to be addressed in the revision of the manuscript. Moreover, the relationships among the sea level changes and the subsidence of the basin, both to a regional and to a local scale have not been analyzed. I suggest perhaps to add in the discussion a short paragraph (half one page) clarifying which are the relationships existing between the oceanographic aspects and the geological processes controlling the sea level fluctuations. This discussion will represent a main added value further improving the quality of the paper. In particular, I think that the relationships between the water column and the height of the sea bottom, as controlled by subsidence, both isostatic and tectonic, need to be clarified. I suggest to the authors to carefully avoid the English grammar repetition and to avoid to be redundant, as it happens in some sections of this manuscript.

Specific comments I suggest to eliminate the quotations of references in the abstract of the paper. Usually, the abstract does not include any quotation. I suggest to put the quotation of references in the paper in a chronological order, not alphabetical one, if not strictly required from the journal. The discussed needs to be expanded taking into account recent literature and geological aspects, as mentioned in the general comments. The conclusions need to be consequently expanded. The captions to figures need to be carefully revised and corrected. Abstract Row 17 …..anthropogenic activity, or estimating the Earth's energy imbalance. Previous authors have estimated the uncertainty…. and have shown that it amounts to….. Row 19 In this study, we extend our previous results providing a comprehensive description of the uncertainties in the satellite GMS record. We analyzed ….. and estimated….ten days. Row 22 Three types of errors have been modeled (drifts, biases, noises) and combined together to derive a realistic estimate of the GMSL error variance-covariance matrix. Rows 23-24 We derived a 90% confidence envelop of the GMSL record on a 10-day basis from the error variance-covariance matrix. Row 25 Then we used a least squared approach …. Row 27 Over 1993-2017 we have found a GMSL trend… Rows 29-30 I suggest to eliminate this sentence. Moreover, in the abstract there is the repetition of the term "estimating". Try to avoid it.

1. Introduction Rows 32-33 The sea level change is a key indicator of global climate change, which integrates changes in several components of the climatic system as a response to climatic variability, both anthropogenic and natural. Rows 39-41 Six research groups (AVISO/CNES, SL_cci/ESA, University of Colorado, CSIRO, NASA/GSFC, NOOA) have processed the sea level raw data provided by satellite altimetry to estimate the GSML series on a 10-day basis (Figure 1) Rows 41-45 There is a repetition of the terms difference and different. Try to re-write this paragraph avoiding the repetition. Row 45 different interpolation methods applied by several groups (Masters et al., 2012; Henry et al., 2014). Rows 45-49 This spread is smaller than the real uncertainty in the sea level trend, because all the research groups have used similar methods and corrections to process the raw data and thus several sources of

systematic uncertainty are not accounted for in the spread. Rows 50-54 In a previous study Ablain et al. (2009) have proposed a realistic estimate of the uncertainty in the GMSL trend over 1993-2008, using an approach based on the error budget. They have identified the radiometer wet tropospheric correction as one of the main sources of error. They have also proposed the orbital determination. . . . . . Row 54 When all the terms were accounted for, they have found . . . .. Rows 56-58 In the framework of the ESA Sea Level Climate Change Initiative (SL_cci), significant improvements have been obtained estimating the sea level from space (Ablain et al., 2015; Quartly et al., 2017; Legeais et al., 2018) to get closer to the GCOS requirements. Rows 61-64 During the second altimetry decade (2002-2014) Ablain et al. (2015) have estimated that the uncertainty of the GMSL trend was lower than. . . . . . Rows 65-67 In previous studies the uncertainty in GMSL has been estimated for long-term trends (periods of 10 years or more, starting in 1993), for inter-annual time scales (between 1 and 5 years) and annual time scales (Ablain et al., 2009; 2015). Rows 67-74 This estimation of the uncertainty at three time-scales is a valuable first step, but it is not enough, as it does not fully meet the needs of the scientific community. In many climatic studies the GMSL uncertainty is required at different time scale and span within the 25-year altimetry record. In sea level budget studies based on the evolution of GMSL components, these estimates have been carried out at monthly time scale. In this way, the GMSL monthly changes have been interpreted in terms of changes of oceanic masses (GRACE mission). Rows 74-80 This is also the case of studies estimating the Earth's energy imbalance with the sea-level budget approach (Meyssignac et al., 2018). In the studies on the detection and the attribution of climate change (Slangen et al., 2017), the uncertainty in the trend estimates is needed, but over different time spans that that ones addressed by Ablain et al. (2009; 2015) and by Legeais et al. (2018). The uncertainty on different metrics is often needed. Dieng et al. (2017) and Nerem et al. (2018) have recently estimated the acceleration in the GMSL over 1993-1997, finding a small acceleration (0.08 mm/yr2) over the 25 year long altimetry record. Rows 79-80 I suggest to eliminate this sentence, it is quite redundant. Rows 81-87 In this paper we focus
on the uncertainty in the GMSL record arising from instrumental errors in the satellite altimetry. The uncertainties of the measurements have been quantified in the GMSL record. This is an important information for the studies in the detection and attribution of the climatic changes, estimating the rise of global mean sea level as a response to the anthropogenic activity. In the detection-attribution studies the response of the GMSL to the anthropogenic activity needs to be separated from that one to climatic variability, representing an additional source of uncertainty. Rows 87-89 I suggest to eliminate these two sentences. They are quite redundant. Rows 98-101 We used an error budget approach to a global scale on a 10 day basis in order to estimate the error variance-covariance matrix. We considered all the major sources of uncertainty in the altimetry measurements, including the wet tropospheric correction, the orbital solutions and the inter-calibration of satellites. We have also taken into account the time correlation between the different sources of uncertainty (section 2). The errors have been separately characterized for each altimetry mission, since they have been affected by different sources of errors (section 2). Rows 105-106 I suggest to eliminate this sentence, it is also very redundant.

1. GMSL data series Rows 110-117 Each group processes the 1-Hz data with geophysical corrections to correct the altimetry measurement for various aliasing, biases and drifts, caused by different atmospheric conditions, sea states, ocean tides and other causes (Ablain et al., 2009). They spatially average the data over each 10-day orbital cycle to provide GMSL estimates on a 10-day basis. The differences among the GMSL estimates from several groups arise from data editing, from difference in the geophysical corrections and from differences in the used method to spatially average the individual measurements during the orbital cycles (Masters et al., 2012; Henry et al., 2014). Rows 117-121 Recently, the comparisons of the GMSL time series derived from satellite altimetry with independent estimates are based on the tide gauge records (Valledeau et al., 2012; Watson et al., 2015) or on the combination of the contribution to the sea level from thermal expansion, land ice melt and land-water storage (Dieng et al., 2017). They have shown that there was a drift in the GMSL record over the

period 1993-1998. This drift is caused by an erroneous onboard calibration correction on TOPEX altimeter side-A (noted TOPEX-A).

2. Altimetry GMSL error budget Rows 138-140 This section describes the different errors that affect the altimetry GMSL record. It builds on the GMSL error budget presented in Ablain et al. (2009) and extends this work taking into account the new altimeter missions (Jason-2, Jason-3) and the recent findings on altimetry error estimates. Row 144 ..... and by the correlation time-scale ($\lambda$). Row 147 Add a point at the end of the sentence. Row 149 The biases can arise .. Row 159 The drifts may occur in the GMSL record.... Rows 163-170 This drift has been corrected by using several empirical approaches (Ablain, 2017; Beckley et al., 2017; Dieng et al., 2017), that are all affected by a significant uncertainty. We estimated this uncertainty to be..... with a comparison between an independent GMSL estimate based on tide gauge records (Ablain, 2017). For the TOPEX-B record, no GMSL drift has been reported, but Ablain et al. (2012) showed significant sigma-0 instabilities in the order of 0.1 dB, which generate through the sea-state bias correction an uncertainty.....(February 1999 – April 2002). Concerning the ITRF realization Couhert et al. (2015) have shown that....... Rows 176-183 The residual time correlated errors are separated into two different groups, depending on their correlation time scales. The first group gathers errors with short correlation time scales, i.e. lower than two months and between two months and one year. The second group gathers errors with long correlation time scales between 5 and 10 years. In the first group the errors are mainly due to the geophysical corrections (ocean tides, atmospheric corrections), to the altimeter corrections (sea-state bias correction, altimeter ionospheric corrections), to the orbital calculation and to the potential altimeter instabilities (altimeter range and sigma-0 instabilities). At time scales below one year, the variability of the corrections' time series is dominated by errors, such that the variance of the error in each ..... Rows 184-186 For errors with correlation time scales lower than 2 months, we estimated the standard deviation ($\sigma$) of the error from the correction's time series filtered with a 2-month high-pass filter. Since the standard deviation of the errors depends on the different altimeter missions, the standard deviation has been separately estimated for each altimeter mission. Rows 196-204 In the second group of residual time correlated errors, the errors are due to the onboard microwave radiometer calibration, yielding instabilities in the wet troposphere correction and also to the orbital calculation (Couhert et al., 2015). Since these errors are correlated at a time scale longer than 5 years, they can not be estimated with the standard deviation of the correction time series, too short (25-year long) to sample the time correlation. For this group of residual-time-correlated errors we used simple models to represent the time correlation of the errors. For the wet troposphere correction, several studies (Legeais et al., 2014; Thao et al., 2014; Fernandes et al., 2015) have identified long-term differences among the computed corrections from the different microwave radiometers and from different atmospheric re-analyses (Dee et al., 2011).

3. The GMSL error variance-covariance matrix Rows 221-222 In this section we derived the error variance-covariance matrix $(\sum) of the GMSL from the error budget described in the section 2. We assumed that all the error sources shown in Table 1 are indepe$

· · ·

Row 231 alongtime Row 234 . . ..but not in the mean time GMSL Row 236 This is the reason because Row 238 For the drifts . . ...takes the shape

4. GMSL uncertainty envelope Row 258 We estimated. . .. . ... Rows 270-275 In Figure 4 we superimposed the GMSL time series (average of the GMSL time series in Figure 1) and the associated uncertainty envelop. For the TOPEX-A period we tested three different curves with three different corrections based on the removal of the Cal-1 mode (Beckley et al., 2017), on the comparison with tide gauges (Watson et al., 2015; Ablain, 2017) or based on a sea level closure budget approach (Dieng et al., 2017). The uncertainty envelop is centered on the record corrected for TOPEX-A drift with the correction based on Ablain et al. (2017). As it has been expected, all the empirically corrected GMSL records are within the uncertainty envelop.

5. Uncertainty in GMSL trend and acceleration Rows 279-281 The variance-covariance

matrix can be used to derive the uncertainty on any metric based on the GMSL time series. In this section we used the error variance-covariance matrix to estimate the uncertainty on the GMSL trend and acceleration over any period of 5 years and more within 1993-2017. Rows 282-287 Recently, several studies (Watson et al., 2015; Dieng et al., 2017; Nerem et al., 2018; WCRP Global Sea Level Budget Group, 2018) have found a significant acceleration in the GMSL record from satellite altimetry (after correction for the TOPEX-A drift). The occurrence of an acceleration in the record should not change the estimation of the trend when calculated with a least squared approach. However, it can affect the estimation of the uncertainty on the trend. To cope with this issue, we address here at the same time both the estimation of the trend and acceleration in the GMSL record. In order to obtain this objective we used a second order polynomial as a predictor. Rows 300-304 The most common method to estimate the GMSL trend and acceleration is the Ordinary Least Squares (OLS) estimator in its classical form (Cazenave and Llovel, 2010; Masters et al., 2012; Dieng et al., 2015; Nerem et al., 2018). This is also the most common method to estimate trends and acceleration in other climate essential variables (Hartmann et al., 2014 and references therein). Rows 311-314 To address this issue, we used a more general formalism to integrate the GMSL error in the trend uncertainty estimation, following Ablain et al. (2009), Ribes et al. (2016) and IPCC AR5 (Hartmann et al., 2014; see in particular Box 2.2 and Supplementary Material). Row 327 Eliminate the space Rows 341-344 and Rows 344-346 Check the English form Rows 354-355 The periods for which the acceleration in sea level is significant at the 90% confidence level are shown in Fig. 8. Rows 362-363 It is unclear which is the relationship between the acceleration of Global Mean Sea Level and the volcanic eruptions (Mount Pinatubo). Rows 364-371 The period for which the trend in sea level is significant at the 90% confidence level is shown in Fig. 9. In periods when the acceleration is not significant, the second order polynomial that we used as a predictor to estimate the trend and the acceleration does not hold anymore in principle. For these periods we should turn out a first order polynomial. The use of a first order polynomial does not affect the trend estimates, but only

the trend uncertainty estimates. We checked for differences in the trend uncertainty when using either second order or first order polynomial predictors. We found that these differences are negligible (not shown). Fig. 9 indicates that for periods of 5 years and longer, the trend in GMSL is always significant at 90% CL over the whole record. At the end of the record the trend tends to increase. This is consistent with the acceleration plot in Figure 6. 6. Conclusions Row 379 . . . . . .measurement also increases and the description of the errors improves. Rows 383-385 The uncertainty of the GMSL here computed shows the reliability of altimetry measurements in order to accurately describe the evolution of the GMSL on all time scales from 10 days to 25 years. It also shows the reliability of altimetry measurements in order to estimate the trends and the accelerations of the sea level. Row 387 . . . . . . .we estimated. . . . . . Rows 391-394 In this study several assumptions have been made, that could be improved in the future. Firstly, the modeling of altimeter errors should be regularly revisited and improved to take into account a better knowledge of errors. . . . . . . . . Concealing the mathematical formalism, the OLS method. . . . . . ..

CAPTIONS TO FIGURES (from January 1993 to December 2017). I suggest to correct in all the captions. Figure 1: Evolution of the GMSL series (corrected for TOPEX-A drift by using Ablain et al., 2009 TOPEX-A correction) from six different groups (AVISO/CNES, CSIRO, University of Colorado, SL_cci/ESA, NASA/GSFC, NOOA). The SL_cci/ESA covers a period from January 1993 to December 2016, while all the other products cover the full 25-year period (from January 1993 to December 2018). Seasonal (annual and semi-annual) signals have been removed and a 6-month smoothing has been applied. An averaged solution has been computed from the six groups. GMSL time series have the same average on the 1993-2015 period (common period) and the averaged solution starts at zero in 1993. The averaged solution without TOPEX-A correction has also been represented. A GIA correction of 0.3 mm/year has been subtracted to each dataset. Figure 2: Error variance-covariance matrix of altimeter GMSL on the 25-years period (from January 1993 to December 2017). Figure 3: Evolution in time of GMSL measurement uncertainty within a 90% confidence level

(1.65 $\sigma$) on the 25-years period (from January 1993 to December 2017). Figure 4: Evolution of the AVISO/GMSL with different TOPEX-A corrections. On the black, red and green curves, the TOPEX-A drift correction has been respectively applied based on Ablain (2017), Watson et al. (2015), Dieng et al. (2017) and Beckley et al. (2017). The uncertainty envelope, as well as the trend and acceleration uncertainties are given to a 90% confidence level (1.65 $\sigma$). Seasonal (annual and semi-annual) signals removed and 6-month smoothing have been applied. GIA correction has also been applied. Figure 5: GMSL trend uncertainties (mm/yr) estimated for all altimeter period within a 25-years period (from January 1993 to December 2017). The confidence level is 90 % (1.65 $\sigma$). Each colored pixel respectively represents the half-size of the 90 % confidence range in the GMSL trend. The values are given in mm/y. The vertical axis indicates the length of the period (ranging from 1 to 25 years) considered in the computation of the trend, while the horizontal axis indicates the center date of the period (for example 2000 for the 20-year period 1990-2009). Figure 8: GMSL acceleration using the AVISO GMSL time series corrected for the TOPEX-A drift using the correction proposed by Ablain (2017): the acceleration in the shaded areas is not significant (lower than the acceleration uncertainties at the 90% confidence level). The length of the window (in years) is represented on the vertical axis and the central date of the used window is represented on the horizontal axis.

Technical corrections Row 288 . . . no observations Row 455 SEA LEVEL BUDGET. Why capitals? Check and correct. Row 478 Marine Geodesy, 35 (suppl. 1), 20-41 Row 503 Marine Geodesy, 35 (suppl. 1), 42-60.

REFERENCES Ablain M., and Coauthors (2015) Improved sea level record over the satellite altimetry era (1993- 2010) from the Climate Change Initiative Project. Ocean Sciences, 11, 67–82, https://doi.org/10.5194/os-11-67-2015. Ablain M., Legeais J.F., Prandi P., Marcos M., Fenoglio-Marc L., Dieng H.B., Benveniste J., Cazenave A. (2017) Satellite altimetry-based sea level at global and regional scales. Surveys in Geophysics, 38, 7–31, https://doi.org/10.1007/s10712-016-9389-8.

Abraham J.P. et al. (2013) A review of global ocean temperature observations: Implications for ocean heat content estimates and climate change. Review in Geophysics, 51: 450–483.

Allan R.P. et al. (2014) Changes in global net radiative imbalance 1985-2012. Geophysical Research Letters, 41, 5588–5597.

Aucan J., Merrifield M.A., Povreau N. (2017) Historical Sea Level in the South Pacific from Rescued Archives, Geodetic Measurements, and Satellite Altimetry. Pure and Applied Geophysics, 174 (10), 3813-3823. Baki Iz H., Shum C.K., Kuo C.Y. (2018) Sea level accelerations at globally distributed tide gauge stations during the satellite altimetry era. J. Geod. Sci. 2018; 8:130–135.

Beckley B.D. et al. (2010) Assessment of the Jason-2 extension to the TOPEX/Poseidon, Jason-1 sea-surface height time series for global mean sea level monitoring. Marine Geodesy, 33 (Suppl 1), 447–471.

Beckley B. D., Callahan P.S., Hancock D.W., Mitchum G.T., Ray R.D. (2017) On the "Cal-Mode"correction to TOPEX satellite altimetry and its effect on the global mean sea level time series. Journal of Geophysical Research (Oceans) 122, 8371–8384. https://doi.org /10.1002/2017JC013090.

Boening C., Willis J.K., Landerer F.W., Nerem R.S., Fasullo J. (2012) The 2011 La Niña: So strong, the oceans fell. Geophysical Research Letters, 39, L19602.

Cazenave A., Dieng H.B., Meyssignac B., Von Schuckmann K., Decharme B., Berthier E. (2014) The rate of sea-level rise. Nat. Climate Change, 4, 358–361, https://doi.org/10.1038/ nclimate2159. Chambers D.P., Wahr J., Tamisiea M.E. , Nerem R.S. (2010) Ocean mass from GRACE and glacial isostatic adjustment. J. Geophys. Res., 115, B11415, https://doi.org/10.1029/2010JB007530.

Chen, X., Zhang X., Church J.A., Watson C.S., King M.A., Monselesan D., Legresy B., Harig C. (2017a) The increasing rate of global mean sea-level rise during 1993–2014.
Natural Climate Change, 7, 492–495, https://doi.org/10.1038/nclimate3325.

Cheng L. et al. (2017b) Improved estimates of ocean heat content from 1960 to 2015. Scientific Advances, 3, e1601545.

Church J. A., White N.J. (2006) A 20th century acceleration in global sea-level rise. Geophysical Research Letters, 33, L01602, https://doi.org/10.1029/2005GL024826.

Church J. A., White N.J. (2011) Sea-level rise from the late 19th to the early 21st century. Surveys in Geophysics, 32, 585–602, https://doi.org/10.1007/s10712-011-9119-1.

Church J.A. et al. (2013) Sea level change. Climate Change 2013: The Physical Science Basis. Contribution of Working Group I to the Fifth Assessment Report of the Intergovernmental Panel on Climate Change (Cambridge Univ Press, Cambridge, UK), pp 1137–1215.

Clark J.A., Primus J.A. (1987) Sea-level changes resulting from future retreat of ice sheets: An effect of $CO_2$ warming of the climate. In: Tooley M.J. and Shennan I. (Eds.), Sea-Level Changes. Institute of British Geographers, 356–370.

Conrad C. P., Hager B.H. (1997) Spatial variations in the rate of sea level rise caused by the present-day melting of glaciers and ice sheets. Geophysical Research Letters, 24, 1503–1506, https:// doi.org/10.1029/97GL01338.

Curry J. (2018) Sea Level and Climate Change. Special Report, 25 November 2018, 79 pp. Dahlen, F. A. (1976) The passive influence of the oceans upon the rotation of the Earth. Geophysical Journal International, 46, 363–406, https:// doi.org/10.1111/j.1365-246X.1976.tb04163.x.

Dangendorf, S., Marcos M., Wöppelmann G., Conrad C.P., Frederikse T., Riva R. (2017) Reassessment of 20th century global mean sea level rise. Proc. Natl. Acad. Sci. USA, 114, 5946–5951, https://doi.org/10.1073/pnas.1616007114.

Davis J. L., Mitrovica J.X. (1996) Glacial isostatic adjustment and the anomalous tide gauge record of eastern North America. Nature, 379, 331–333, https://doi.org/10.1038/379331a0.

Desai S., Wahr J., Beckley B. (2015) Revisiting the pole tide for and from satellite altimetry. J. Geod., 89, 1233–1243, https:// doi.org/10.1007/s00190-015-0848-7.

Desbruyères D., McDonagh E.L., King B.A. (2016) Observational advances in estimates of oceanic heating. Current Climate Change Reports, 2,127–134.

Dieng H. B., Cazenave A., Meyssignac B., Ablain M. (2017) New estimate of the current rate of sea level rise from a sea level budget approach. Geophysical Research Letters, 44, 3744–3751, https://doi.org/10.1002/2017GL073308.

Esselborn S., Rudenko S., Schon T. (2018) Orbit-related sea level errors for TOPEX altimetry at seasonal to decadal timescales. Ocean Science, 14, 205-223.

Farrell W. E., Clark J.A. (1976) On postglacial sea level. Geophysical Journal International, 46, 647–667, https://doi.org/10.1111/ j.1365-246X.1976.tb01252.x.

Fasullo J.T., Boening C., Landerer F.W., Nerem R.S (2013) Australia's unique influence on global sea level in 2010–2011. Geophysical Research Letters, 40, 4368–4373.

Fasullo J.T., Nerem R.S., Hamlington B.(2016) Is the detection of accelerated sea level rise imminent? Scientific Reports 6: 31245, doi 10:1038/srep31245.

Frederikse T., Jevrejeva S., Riva R.E.M., Dangendorf S. (2017a) A consistent sea-level reconstruction and its budget on basin and global scales over 1958–2014. Journal of Climate, 31, 1267– 1280, https://doi.org/10.1175/JCLI-D-17-0502.1.

Frederikse T., Riva R.E.M., King M.A. (2017b) Ocean bottom deformation due to present-day mass redistribution and its impact on sea level observations. Geophysical Research Letters, 44, 12 306–12 314, https://doi.org/10.1002/2017GL075419.

Frederikse T., Jevrejeva S., Riva R.E.M., Dangendorf S. (2018) A Consistent Sea-Level Reconstruction and Its Budget on Basin and Global Scales over 1958–2014. Journal

of Climate, 31, 1267-1280.

Gardner A.S. et al. (2013) A reconciled estimate of glacier contributions to sea level rise: 2003 to 2009. Science, 340, 852–857.

Gornitz V., Lin N., Oppenheimer M., Horton R., Kopp R., Bader D. (2019) New York City Panel on Climate Change 2019 Report Chapter 3: Sea Level Rise. Annals of the New York Academy of Sciences, Special Issue Advancing Tools and Methods for Flexible Adaptation Pathways and Science Policy Integration. Annals of the New York Academy of Sciences, New York Panel on Climate Change, ISSN 0077-8923, 94 pp.

Gregory, J. M., and Coauthors (2013) Twentieth-century global mean sea level rise: Is the whole greater than the sum of the parts? Journal of Climate, 26, 4476–4499, https://doi.org/10.1175/ JCLI-D-12-00319.1.

Haigh ID, et al. (2014) Timescales for detecting a significant acceleration in sea level rise. Nature Communications, 5:3635.

Hamlington B. D., Thompson P.R. (2015) Considerations for estimating the 20th century trend in global mean sea level. Geophysical Research Letters, 42, 4102–4109, https://doi.org/10.1002/2015GL064177.

Hamlington B.D., Leben R.R., Strassburg M.W., Nerem R.S., Kim K.Y. (2013) Contribution of the pacific decadal oscillation to global mean sea level trends. Geophysical Research Letters, 40, 5171–5175.

Hamlington B.D. et al. (2016) An ongoing shift in Pacific Ocean sea level. Journal of Geophysical Research, Oceans, 121, 5084–5097.

Hamlington B.D, Reager J.T., Lo M.H, Karnauskas K.B, Leben R.R (2017) Separating decadal global water cycle variability from sea level rise. Scientific Reports, 7:995.

Handoko E.Y., Hariyadi H. (2018) Merged envisat and jason satellite altimeters using crossovers adjustment to determine sea level variability. IOP Conference Series, Earth
Environmental Sciences, 200, 012035.

Hay C. C., Morrow E., Kopp R.E., Mitrovica J.X. (2015) Probabilistic re-analysis of twentieth century sea-level rise. Nature, 517, 481–484, https://doi.org/10.1038/nature14093.

Herring, S. C., Christidis N., Hoell A., Hoerling M.P., Stott P.A., Eds. (2019) Explaining Extreme Events of 2017 from a Climate Perspective. Bull. Amer. Meteor. Soc., 100 (1), S1–S117, https://doi.org/10.1175/BAMS-ExplainingExtremeEvents2017.1.

Kay J.E., et al. (2014) The community earth system model (CESM) large ensemble project: A community resource for studying climate change in the presence of internal climate variability. Bulletin of American Meteorological Society, 96, 1333–1349.

Kendall R. A., Mitrovica J.X., Milne G.A. (2005) On post- glacial sea level—II. Numerical formulation and comparative results on spherically symmetric models. Geophysical Journal International, 161, 679–706, https://doi.org/10.1111/j.1365-246X.2005.02553.x.

Kidwell, D. M., Dietrich J.C., Hagen S.C., Medeiros S.C. (2017) An Earth's Future Special Collection: Impacts of the coastal dynamics of sea level rise on low-gradient coastal landscapes. Earth's Future, 5,2–9, doi:10.1002/2016EF000493.

Jevrejeva S., Moore J.C., Grinsted A., Woodworth P.L. (2008) Recent global sea level acceleration started over 200 years ago? Geophysical Research Letters, 35, L08715, https://doi.org/10.1029/ 2008GL033611.

Jevrejeva S., Matthews A., Slangen A. (2017) The twentieth-century sea level budget: Recent progress and challenges. Surveys in Geophysics, 38, 295–307, https://doi.org/10.1007/s10712-016-9405-z.

Johnson G.C., Lyman J.M., Loeb N.G. (2016) Improving estimates of Earth's energy imbalance. Natural Climate Change, 6, 639–640.
Johnston P. (1993) The effect of spatially non-uniform water loads on predictions of sea-level change. Geophysical Journal International, 114, 615– 634, https://doi.org/10.1111/j.1365 246X.1993.tb06992.x.

Lickley M. J., Hay C.C., Tamisiea M.E., Mitrovica J.X. (2018) Bias in Estimates of Global Mean Sea Level Change Inferred from Satellite Altimetry. Journal of Climate, 31, 5263-5271.

Melachroinos S.A., Lemoine F.G., Zelensky N.P., Rowlands D.D., Luthcke S.B., Bordyugov O. (2013) The effect of geocenter motion on Jason-2 orbits and the mean sea level. Advances in Space Research, 51, 1323–1334, https://doi.org/10.1016/j.asr.2012.06.004.

Merrifield M.A, Merrifield S.T, Mitchum G.T (2009) An anomalous recent acceleration of global sea level rise. Journal of Climate, 22, 5772–5781.

Milne G. A., Mitrovica J.X. (1996) Postglacial sea-level change on a rotating Earth: First results from a gravitationally self-consistent sea-level equation. Geophysical Journal International, 126, F13–F20, https://doi.org/10.1111/j.1365-246X.1996.tb04691.x.

Mitchum G.T (2000) An improved calibration of satellite altimetric heights using tide gauge sea levels with adjustment for land motion. Marine Geodesy, 23, 145–166.

Mitrovica J. X., Milne G. A. (2003) On post-glacial sea level: I. General theory. Geophysical Journal International, 154, 253–267, https://doi.org/ 10.1046/j.1365-246X.2003.01942.x.

Mitrovica J., Tamisiea M.E., Davis J.L., Milne G.A. (2001) Recent mass balance of polar ice sheets from patterns of global sea-level change. Nature, 409, 1026–1029, https://doi.org/10.1038/35059054.

Nerem R.S., Fasullo J. (2018) Observations of the rate and acceleration of global mean sea level change. American Metereological Society, December 2018, BAMS, S1-S3.

Nerem R.S, Chambers D.P, Leuliette E.W, Mitchum G.T, Giese B.S (1999) Variations in global mean sea level associated with the 1997–1998 ENSO event: Implications for measuring long term sea level change. Geophysical Research Letters, 26, 3005–3008.

Nerem R. S., Chambers D.P., Choe C., Mitchum G.T. (2010) Estimating mean sea level change from the TOPEX and Jason altimeter missions. Marine Geodesy, 33, 435–446, https://doi.org/ 10.1080/01490419.2010.491031.

Nerem R.S., Beckley B.D., Fasullo J.T., Hamlington B.D., Masters D., Mitchum J.T. (2018) Climate-change–driven accelerated sea-level rise detected in the altimeter era. Proceedings National Acad. Sci. USA, 115 (9), 2022–2025, https://doi.org/10.1073/pnas.1717312115.

Prandi P., Cazenave A. Becker M. (2009) Is coastal mean sea level rising faster than the global mean? A comparison between tide gauges and satellite altimetry over 1993–2007. Geophysical Research Letters, 36, L05602, https://doi.org/10.1029/2008GL036564.

Ray R. D., Luthcke S.B., van Dam T. (2013) Monthly crustal loading corrections for satellite altimetry. J. Atmos. Oceanic Technol., 30, 999–1005, https://doi.org/10.1175/JTECH-D-12-00152.1.

Shepherd A., and Coauthors (2018) Mass balance of the Antarctic Ice Sheet from 1992 to 2017. Nature, 558, 219–222, https://doi.org/10.1038/s41586-018-0179-y.

Slangen A. B. A., Church J.A., Agosta C., Fettweis X., Marzeion B., Richter K. (2016) Anthropogenic forcing dominates global mean sea-level rise since 1970. Natural Climate Change, 6, 701–705, https://doi .org/10.1038/ncl i mate2991.

Slangen A.B.A., Church J.A. Melet A., Meyssignac B., Fettweis X., Palmer M.D., Agosta C., Ligtenberg S.R.M., Richter K., Spada G., Champollion N., Marzeion B., Roberts C.D. (2017) Evaluating Model Simulations of Twentieth-Century Sea Level Rise. Part I: Global Mean Sea Level Change. Journal of Climate, 30, 8359-8563.

Swart N. C., Fyfe J.C., Hawkins E., Kay J.E. Jahn A. (2015) Influence of internal variability on Arctic sea ice trends. Natural Climate Change, 5, 86–89, https://doi.org/10.1038/nclimate2483.

Spada G. (2017) Glacial isostatic adjustment and contemporary sea level rise: An overview. Survey in Geophysics, 38, 153–185, https:// doi.org/10.1007/s10712-016-9379-x.

Spada G., Galassi G. (2016) Spectral analysis of sea level during the altimetry era, and evidence for GIA and glacial melting fingerprints. Global and Planetary Change, 143, 34–49, https://doi.org/ 10.1016/j.gloplacha.2016.05.006.

Tamisiea M. E. (2011) Ongoing glacial isostatic contributions to observations of sea level change. Geophysical Journal International, 186, 1036– 1044, https://doi.org/10.1111/j.1365 246X.2011.05116.x.

Thompson P. R ., Hamlington B.D., Landerer F. W., Adhikari S. (2016) Are long tide gauge records in the wrong place to measure global mean sea level rise? Geophysical Research Letters, 43, 10 403–10 411, https:// doi.org/10.1002/2016GL070552.

Trenberth K.E., Fasullo J.T., von Schuckmann K., Cheng L. (2016) Insights into Earth's energy imbalance from multiple sources. Journal of Climate, 29, 7495–7505.

Vaughan D. G. and Coauthors (2013) Observations: Cryosphere. Climate Change 2013: The Physical Science Basis, T. F. Stocker et al., Eds., Cambridge University Press, 317–382.

Wahr, J., R. S. Nerem, and S. V. Bettadpur, 2015: The pole tide and its effect on GRACE time-variable gravity measurements: Implications for estimates of surface mass variations. J. Geophys. Res. Solid Earth, 120, 4597–4615, https://doi.org/10.1002/2015JB011986.

Wang G., Cheng L., Abraham J., Li C. (2017) Consensuses and discrepancies of basin-scale ocean heat content changes in different ocean analyses. Climate Dynamics,

10.1007/s00382017-3751-5.

Watkins M.M., Wiese D.N., Yuan D.N., Boening C., Landerer F.W. (2015) Improved methods for observing Earth's time variable mass distribution with GRACE using spherical cap mascons. Journal of Geophysical Research, Solid Earth, 120, 2648–2671.

Watson C. S., White N.J. , Church J.A., King M.A., Burgette R.J., Legresy B. (2015) Unabated global mean sea-level rise over the satellite altimeter era. Natural Climate Change, 5, 565–568, https://doi.org/10.1038/ nclimate2635.

Wiese D.N., Yuan D.N., Boening C., Landerer F.W., Watkins M.M. (2016) JPL GRACE Mascon Ocean, Ice, and Hydrology Equivalent Water Height RL05M.1 CRI Filtered (PO.DAAC, Pasadena, CA). Version 2. Available at dx.doi.org/10.5067/TEMSC-2LCR5. Accessed October 1, 2017.

Wouters B., Bamber J.L., van den Broeke M.R., Lenaerts J.T.M., Sasgen I. (2013) Limits in detecting acceleration of ice sheet mass loss due to climate variability. Natural Geosciences, 6, 613–616.

Please also note the supplement to this comment:
https://www.earth-syst-sci-data-discuss.net/essd-2019-10/essd-2019-10-RC2-supplement.pdf

―――――――――――――――

---

## Author Comment (AC1) · 10 Jul 2019

RC1 : This paper provides the first estimate of an error variance-covariance matrix for altimeter measurements of global mean sea level rise. The authors then derive a 90% confidence interval of GMSL on a 10-day basis and estimate the trend and acceleration of GMSL over 5 year or longer intervals. Overall the paper is easy to understand and could potentially provide a useful quantification of uncertainty. However, my primary concern is with the treatment of GIA uncertainty and the authors must address this.

Answer to RC1: We thank reviewer 1 for this positive review. In the revised manuscript and the detailed response below we now address reviewer 1's concern about the treatment of the GRD correction associated to present day mass loss. We thank reviewer

1 for pointing us to this flaw in the manuscript.

RC1 : The authors note that they use the Spada 2017 estimate of 0.05 mm/year for GIA uncertainty. This uncertainty estimate is for the GIA component due to the ongoing changes in the Earth's crust since the last glacial maximum (LGM) but does not include modern day melt contributions to GIA. As the authors are aware, the LGM-GIA response is typically accounted for in altimeter-based estimates of GMSL by adding 0.3 mm/yr to the altimeter-derived estimate of GMSL. However, this estimate does not account for deformations of the ocean bottom due to modern melt, which can introduce biases in both the mean trend and acceleration term. See, for example, Frederikse et al. 2017 and Lickley et al. 2018. This correction need not be included if the authors wish to use altimeter measurements to estimate changes in sea surface height instead of sea level. However, the authors explicitly reference estimates of changes in sea level (lines 117- 120) where they compare altimeter estimates of GMSL to changes in ocean volume as measured by tide gauges, or the sum of the contributions to changes in ocean volume. To be consistent, I believe this additional source of GIA uncertainty should be accounted for. Alternatively, they could remove the GIA estimate altogether and state upfront that this is an estimate of the uncertainty in sea surface height and cannot be compared to volumetric changes in sea level.

Answer to RC1: Reviewer 1 is right, we need to include the Frederikse et al. (2017) and Lickley et al. (2018) elastic correction in our study because we compare altimeter estimates of GMSL rise to changes in ocean volume as measured by tide gauges. We now correct our estimate of the GMSL rise by +0.10 mm/yr (in the text and in figures 1, 4 and 9) as recommended by Frederikse et al. (2017). The uncertainty in this correction arises mainly from uncertainty associated to the procedure to solve the sea level equation, uncertainty in the choice of the Love numbers, uncertainty generated by the truncation degree of the spherical harmonics and the uncertainty in the mass redistribution. Because the elastic response of the Earth and its main parameters (i.e. the sea level equation, the Love numbers, the spherical harmonic development) are

reasonably well defined (Mitrovica et al., 2011), the uncertainty in this correction is largely dominated by uncertainties in the mass redistribution (Frederikse et al. 2017). The uncertainty on the mass redistribution is about $\pm 10\%$ on the current ice mass loss (e.g. Blazquez et al. 2018, The WCRP sea level budget group 2018). Since the elastic response of the solid Earth is linear , the uncertainty in the ocean bottom motion associated to the uncertainty in the mass redistribution should also amount $\pm 10\%$ of the total correction. It yields an uncertainty of $\pm 0.01$ mm/yr on the elastic correction. This uncertainty is very small. It is an order of magnitude smaller than the uncertainty considered in this study (see Table 1). So we neglect this source of uncertainty in our study. We now write a paragraph on line 335 to explain this.

RC1 : Specific Comments:There are a number of grammatical errors and issues with vocabulary choice through-out. Please check! Here are a few examples: Line 41: add an s to "altimeter" Replace "confidence enve-lope" with "confidence interval"

Answer to RC1: corrected

RC1 : throughout Line 86: replace "the GMSL" with "GMSL".Line 96: Add "us" after "enables" and remove the "s" on "metrics"

Answer to RC1: corrected

RC1 : Other issues: Line 307, should be 'âĹij' not '=' Please label axes on Figure 5 and 9.

Answer to RC1: corrected

Please also note the supplement to this comment:
https://www.earth-syst-sci-data-discuss.net/essd-2019-10/essd-2019-10-AC1-supplement.pdf

---

## Author Comment (AC2) · 10 Jul 2019

RC2: In this paper the uncertainty in the satellite estimate of Global Mean Sea Level changes, particularly referring to the trend and the acceleration has been evaluated. I have read it with attention, finding that its quality is quite good, in my opinion. The English form is generally good but at some sections it needs to be further improved. Moreover, the research group appears to be qualified in the field of satellite oceanography. Nonetheless this, a moderate revision is still necessary for a further improvement of the paper's quality (see specific comments).

Answer to RC2: We thank reviewer 2 for this positive review. In the revised manuscript and the detailed response below we now address the miswording. We thank reviewer

2 for the detailed reading of our manuscript and for the rewording suggestions.

RC2: The topic of Global Mean Sea Level and its relationships with climate changes has been deeply studied in marine geophysics and satellite oceanography (Ablain et al., 2015; 2017; Abraham et al., 2013; Allan et al., 2014; Aucan et al., 2017; Baki Iz et al., 2018; Beckley et al., 2010; 2017; Boening et al., 2012; Cazenave et al., 2014; Chambers et al., 2010; Chen et al., 2017a; 2017b; Church and White, 2006; 2011; Church et al., 2013; Clark and Primus, 1987; Conrad and Hager, 1997; Curry, 2018; Dahlen, 1976; Dangendorf et al., 2017; Davis and Mitrovica, 1996; Desai et al., 2015; Desbruyeres et al., 2016; Dieng et al., 2017; Esselborn et al., 2018; Farrell and Clark, 1976; Fasullo et al., 2013; 2016; Frederikse et al., 2017a; 2017b; 2018; Gardner et al., 2013; Gornitz et al., 2019; Gregory et al., 2013; Haigh et al., 2014; Hamlington and Thompson, 2015; Hamlington et al., 2013; 2016; 2017; Handoko and Hariyadi, 2018; Hay et al., 2015; Herring et al., 2019; Kay et al., 2014; Kendall et al., 2005; Kidwell et al., 2017; Lickley et al., 2018; Melachroinos et al., 2013; Merrifield et al., 2009; Milne and Mitrovica, 1996; Mitchum, 2000; Mitrovica and Milne, 2003; Mitrovica et al., 2001; Nerem and Fasullo, 2018; Nerem et al., 1999; 2010; 2018; Prandi et al., 2009; Ray et al., 2013; Shepherd et al., 2018; Slangen et al., 2016; 2017; Swart et al., 2015; Spada, 2017; Spada and Galassi, 2016; Tamisiea, 2011; Thompson et al., 2016; Trenberth et al., 2016; Vaughan et al., 2013; Wahr et al., 2015; Wang et al., 2017; Watkins et al., 2015; Watson et al., 2015; Wiese et al., 2016; Wouters et al., 2013). Due to the exceptional abundance of recent scientific literature addressing this research topic, I suggest perhaps to the authors to expand the discussion of their results, taking into account some of the scientific papers listed in the attached references, which have not considered in detail. This could be a general issue to be addressed in the revision of the manuscript.

Answer to RC2: We thank reviewer 2 for suggesting all these publications. This abundant literature address many different scientific questions such as 1) general climate variability (Herring et al., 2019; Kay et al., 2014; Swart et al., 2015) 2) coastal sea

level (Kidwell et al., 2017, Prandi et al., 2009;) 3) the closure of the sea level budget (Boening et al., 2012; Cazenave et al., 2014, Chen et al., 2017a, Dieng et al., 2017, Fasullo et al., 2013, Watson et al., 2015) 4) the 20th century sea level changes (Aucan et al., 2017, Church and White, 2006; 2011, Dangendorf et al., 2017 Frederikse et al., 2017a,2018 Gregory et al., 2013, Hamlington and Thompson, 2015, Hay et al., 2015, Ray et al., 2013, Slangen et al. 2017, Thompson et al., 2016) 5) the contributions to sea level change(Abraham et al., 2013, Chambers et al., 2010, Cheng et al. 2017b , Gardner et al., 2013, Desbruyeres et al., 2016;, Shepherd et al., 2018, Wiese et al., 2016; Conrad and Hager, 1997; Hamlington et al., 2013; 2016; 2017; Nerem et al., 1999, Wang et al., 2017; Watkins et al., 2015, , Wouters et al., 2013) 6) GIA (Milne and Mitrovica, 1996, Kendall et al., 2005, Farrell and Clark, 1976; Mitrovica and Milne, 2003; Mitrovica et al., 2001, Tamisiea, 2011) 7) the acceleration in sea level during the altimetry period (Fasullo et al. 2016;Haigh et al., 2014, Nerem and Fasullo, 2018; Nerem et al. 2018) 8) the topex correction (Beckley et al.; 2017) 9) the altimetry corrections (; Esselborn et al., 2018; , Dahlen, 1976; , Desai et al., 2015, Frederikse et al., 2017a; 2017b, Lickley et al., 2018; Melachroinos et al., 2013; Spada, 2017; Spada and Galassi, 2016; Tamisiea, 2011; Wahr et al., 2015;) 10) the detection and attribution of sea level changes (Slangen et al., 2016; ) 11) the earth energy imbalance (Trenberth et al., 2016; Allan et al., 2014;) 12) sea level from tide gauge records (Davis and Mitrovica, 1996; , Baki Iz et al., 2018, Merrifield et al., 2009, Mitchum, 2000;) 13) sea level projections(Clark and Primus, 1987;) 14) the buiding of a satellite altimetry record (Ablain et al., 2015; 2017; Handoko and Hariyadi, 2018; Nerem et al., 1999; 2010; Beckley et al., 2010) 15) and general overviews on sea level science (Church et al., 2013; Curry, 2018; Gornitz et al., 2019; Vaughan et al., 2013;) We want to highlight here that this paper focuses on the uncertainties in sea level estimates from satellite altimetry. As such only a few of these publications are actually relevant for our purpose . Those are the one related to the scientific questions number 7 and 8. We now consider these publications and include them in our manuscript (except for HAigh et al. 2014 and Nerem and Fasullo 2018 which adress the question of the acceleration in

the sea level response to GHG emissions while we address in our paper the question of sea level changes in reponse to any forcing and to internal variability. As such these two publications are not relevant to our paper). We thank reviewer 2 for pointing these missing references.

RC2: Moreover, the relationships among the sea level changes and the subsidence of the basin, both to a regional and to a local scale have not been analyzed. I suggest perhaps to add in the discussion a short paragraph (half one page) clarifying which are the relationships existing between the oceanographic aspects and the geological processes controlling the sea level fluctuations. This discussion will represent a main added value further improving the quality of the paper. In particular, I think that the relationships between the water column and the height of the sea bottom, as controlled by subsidence, both isostatic and tectonic, need to be clarified.

Answer to RC2: This is done now on line 365 (see revised revision)

RC2: I suggest to the authors to carefully avoid the English grammar repetition and to avoid to be redundant, as it happens in some sections of this manuscript.

Answer to RC2: We have the English grammar mistakes in the revised revision. Please see the specific comments below

Specific comments

RC2: I suggest to eliminate the quotations of references in the abstract of the paper. Usually, the abstract does not include any quotation. Answer to RC2: corrected

RC2: I suggest to put the quotation of references in the paper in a chronological order, not alphabetical one, if not strictly required from the journal. Answer to RC2: This is not possible as the journal requires an alphabetical order

RC2: The discussed needs to be expanded taking into account recent literature and geological aspects, as mentioned in the general comments. The conclusions need to be consequently expanded. Answer to RC2: Please see our answer to the general

comments above

RC2: The captions to figures need to be carefully revised and corrected. Answer to RC2: Done as suggested by reviewer 2 in his specific comments. Please see our answer to the specific comments

Please also note the supplement to this comment:
https://www.earth-syst-sci-data-discuss.net/essd-2019-10/essd-2019-10-AC2-supplement.pdf

**Supplement:**

General comments
In this paper the uncertainty in the satellite estimate of Global Mean Sea Level changes, particularly referring to the trend and the acceleration has been evaluated. I have read it with attention, finding that its quality is quite good, in my opinion. The English form is generally good but at some sections it needs to be further improved. Moreover, the research group appears to be qualified in the field of satellite oceanography. Nonetheless this, a moderate revision is still necessary for a further improvement of the paper's quality (see specific comments).

*We thank reviewer 2 for this positive review. In the revised manuscript and the detailed response below we now address the miswording. We thank reviewer 2 for the detailed reading of our manuscript and for the rewording suggestions.*

The topic of Global Mean Sea Level and its relationships with climate changes has been deeply studied in marine geophysics and satellite oceanography (Ablain et al., 2015; 2017; Abraham et al., 2013; Allan et al., 2014; Aucan et al., 2017; Baki Iz et al., 2018; Beckley et al., 2010; 2017; Boening et al., 2012; Cazenave et al., 2014; Chambers et al., 2010; Chen et al., 2017a; 2017b; Church and White, 2006; 2011; Church et al., 2013; Clark and Primus, 1987; Conrad and Hager, 1997; Curry, 2018; Dahlen, 1976; Dangendorf et al., 2017; Davis and Mitrovica, 1996; Desai et al., 2015; Desbruyeres et al., 2016; Dieng et al., 2017; Esselborn et al., 2018; Farrell and Clark, 1976; Fasullo et al., 2013; 2016; Frederikse et al., 2017a; 2017b; 2018; Gardner et al., 2013; Gornitz et al., 2019; Gregory et al., 2013; Haigh et al., 2014; Hamlington and Thompson, 2015; Hamlington et al., 2013; 2016; 2017; Handoko and Hariyadi, 2018; Hay et al., 2015; Herring et al., 2019; Kay et al., 2014; Kendall et al., 2005; Kidwell et al., 2017; Lickley et al., 2018; Melachroinos et al., 2013; Merrifield et al., 2009; Milne and Mitrovica, 1996; Mitchum, 2000; Mitrovica and Milne, 2003; Mitrovica et al., 2001; Nerem and Fasullo, 2018; Nerem et al., 1999; 2010; 2018; Prandi et al., 2009; Ray et al., 2013; Shepherd et al., 2018; Slangen et al., 2016; 2017; Swart et al., 2015; Spada, 2017; Spada and Galassi, 2016; Tamisiea, 2011; Thompson et al., 2016; Trenberth et al., 2016; Vaughan et al., 2013; Wahr et al., 2015; Wang et al., 2017; Watkins et al., 2015; Watson et al., 2015; Wiese et al., 2016; Wouters et al., 2013).
Due to the exceptional abundance of recent scientific literature addressing this research topic, I suggest perhaps to the authors to expand the discussion of their results, taking into account some of the scientific papers listed in the attached references, which have not considered in detail. This could be a general issue to be addressed in the revision of the manuscript.

*We thank reviewer 2 for suggesting all these publications. This abundant literature address many different scientific questions such as*
1) *general climate variability (Herring et al., 2019; Kay et al., 2014; Swart et al., 2015)*
2) *coastal sea level (Kidwell et al., 2017, Prandi et al., 2009;)*
3) *the closure of the sea level budget (Boening et al., 2012; Cazenave et al., 2014, Chen et al., 2017a, Dieng et al., 2017, Fasullo et al., 2013, Watson et al., 2015)*

4) the 20th century sea level changes (Aucan et al., 2017, Church and White, 2006; 2011, Dangendorf et al., 2017 Frederikse et al., 2017a,2018 Gregory et al., 2013, Hamlington and Thompson, 2015, Hay et al., 2015, Ray et al., 2013, Slangen et al. 2017, Thompson et al., 2016)

5) the contributions to sea level change(Abraham et al., 2013, Chambers et al., 2010, Cheng et al. 2017b , Gardner et al., 2013, Desbruyeres et al., 2016;, Shepherd et al., 2018, Wiese et al., 2016; Conrad and Hager, 1997; Hamlington et al., 2013; 2016; 2017; Nerem et al., 1999, Wang et al., 2017; Watkins et al., 2015, , Wouters et al., 2013)

6) GIA (Milne and Mitrovica, 1996, Kendall et al., 2005, Farrell and Clark, 1976; Mitrovica and Milne, 2003; Mitrovica et al., 2001, Tamisiea, 2011)

7) the acceleration in sea level during the altimetry period (Fasullo et al. 2016;Haigh et al., 2014, Nerem and Fasullo, 2018; Nerem et al. 2018)

8) the topex correction (Beckley et al.; 2017)

9) the altimetry corrections (; Esselborn et al., 2018; , Dahlen, 1976; , Desai et al., 2015, Frederikse et al., 2017a; 2017b, Lickley et al., 2018; Melachroinos et al., 2013; Spada, 2017; Spada and Galassi, 2016; Tamisiea, 2011; Wahr et al., 2015;)

10) the detection and attribution of sea level changes (Slangen et al., 2016; )

11) the earth energy imbalance (Trenberth et al., 2016; Allan et al., 2014;)

12)sea level from tide gauge records (Davis and Mitrovica, 1996; , Baki Iz et al., 2018, Merrifield et al., 2009, Mitchum, 2000;)

13)sea level projections(Clark and Primus, 1987;)

14) the buiding of a satellite altimetry record (Ablain et al., 2015; 2017; Handoko and Hariyadi, 2018; Nerem et al., 1999; 2010; Beckley et al., 2010)

15) and general overviews on sea level science (Church et al., 2013; Curry, 2018; Gornitz et al., 2019; Vaughan et al., 2013;)

We want to highlight here that this paper focuses on the uncertainties in sea level estimates from satellite altimetry. As such only a few of these publications are actually relevant for our purpose . Those are the one related to the scientific questions number 7 and 8. We now consider these publications and include them in our manuscript (except for HAigh et al. 2014 and Nerem and Fasullo 2018  which adress the question of the acceleration in the sea level response to GHG emissions while we address in our paper the question of sea level changes in reponse to any forcing and to internal variability. As such these two publications are not relevant to our paper). We thank reviewer 2 for pointing these missing references.

Moreover, the relationships among the sea level changes and the subsidence of the basin, both to a regional and to a local scale have not been analyzed. I suggest perhaps to add in the discussion a short paragraph (half one page) clarifying which are the relationships existing between the oceanographic aspects and the geological processes controlling the sea level fluctuations. This discussion will represent a main added value further improving the quality of the paper. In particular, I think that the relationships between the water column and the height of the sea bottom, as controlled by subsidence, both isostatic and tectonic, need to be clarified.

*This is done now on line 365*

I suggest to the authors to carefully avoid the English grammar repetition and to avoid to be redundant, as it happens in some sections of this manuscript.

*We have the English grammar mistakes in the revised revision. Please see the specific comments below*

Specific comments
I suggest to eliminate the quotations of references in the abstract of the paper. Usually, the abstract does not include any quotation.
*corrected*
I suggest to put the quotation of references in the paper in a chronological order, not alphabetical one, if not strictly required from the journal.
*This is not possible as the journal requires an alphabetical order*

The discussed needs to be expanded taking into account recent literature and geological aspects, as mentioned in the general comments.
The conclusions need to be consequently expanded.
*Please see our answer to the general comments above*

The captions to figures need to be carefully revised and corrected.
*Done as suggested by reviewer 2 in his specific comments. Please see our answer to the specific comments*

Abstract
Row 17
…..anthropogenic activity, or estimating the Earth's energy imbalance. Previous authors have estimated the uncertainty…. and have shown that it amounts to…..
*corrected*
Row 19
In this study, we extend our previous results providing a comprehensive description of the uncertainties in the satellite GMS record. We analyzed ….. and estimated….ten days.
*corrected*
Row 22
Three types of errors have been modeled (drifts, biases, noises) and combined together to derive a realistic estimate of the GMSL error variance-covariance matrix.
*corrected*
Rows 23-24
We derived a 90% confidence envelop of the GMSL record on a 10-day basis from the error variance-covariance matrix.
*corrected*
Row 25
Then we used a least squared approach ….
*corrected*
Row 27
Over 1993-2017 we have found a GMSL trend…
*corrected*
Rows 29-30
I suggest to eliminate this sentence.
Moreover, in the abstract there is the repetition of the term "estimating". Try to avoid it.
*corrected*

1. Introduction

Rows 32-33

The sea level change is a key indicator of global climate change, which integrates changes in several components of the climatic system as a response to climatic variability, both anthropogenic and natural.

*corrected*

Rows 39-41

Six research groups (AVISO/CNES, SL_cci/ESA, University of Colorado, CSIRO, NASA/GSFC, NOOA) have processed the sea level raw data provided by satellite altimetry to estimate the GSML series on a 10-day basis (Figure 1)

*corrected*

Rows 41-45

There is a repetition of the terms difference and different. Try to re-write this paragraph avoiding the repetition.

*corrected*

Row 45

different interpolation methods applied by several groups (Masters et al., 2012; Henry et al., 2014).

*corrected*

Rows 45-49

This spread is smaller than the real uncertainty in the sea level trend, because all the research groups have used similar methods and corrections to process the raw data and thus several sources of systematic uncertainty are not accounted for in the spread.

*corrected*

Rows 50-54

In a previous study Ablain et al. (2009) have proposed a realistic estimate of the uncertainty in the GMSL trend over 1993-2008, using an approach based on the error budget. They have identified the radiometer wet tropospheric correction as one of the main sources of error. They have also proposed the orbital determination……

*corrected*

Row 54

When all the terms were accounted for, they have found …..

*corrected*

Rows 56-58

In the framework of the ESA Sea Level Climate Change Initiative (SL_cci), significant improvements have been obtained estimating the sea level from space (Ablain et al., 2015; Quartly et al., 2017; Legeais et al., 2018) to get closer to the GCOS requirements.

*corrected*

Rows 61-64

During the second altimetry decade (2002-2014) Ablain et al. (2015) have estimated that the uncertainty of the GMSL trend was lower than……

*corrected*

Rows 65-67

In previous studies the uncertainty in GMSL has been estimated for long-term trends (periods of 10 years or more, starting in 1993), for inter-annual time scales (between 1 and 5 years) and annual time scales (Ablain et al., 2009; 2015).

*corrected*

Rows 67-74

This estimation of the uncertainty at three time-scales is a valuable first step, but it is not enough, as it does not fully meet the needs of the scientific community. In many climatic studies the GMSL uncertainty is required at different time scale and span within the 25-year altimetry record. In sea level budget studies based on the evolution of GMSL components, these estimates have been carried out at monthly time scale. In this way, the GMSL monthly changes have been interpreted in terms of changes of oceanic masses (GRACE mission).

*corrected*

Rows 74-80

This is also the case of studies estimating the Earth's energy imbalance with the sea-level budget approach (Meyssignac et al., 2018). In the studies on the detection and the attribution of climate change (Slangen et al., 2017), the uncertainty in the trend estimates is needed, but over different time spans that that ones addressed by Ablain et al. (2009; 2015) and by Legeais et al. (2018). The uncertainty on different metrics is often needed. Dieng et al. (2017) and Nerem et al. (2018) have recently estimated the acceleration in the GMSL over 1993-1997, finding a small acceleration (0.08 mm/yr2) over the 25 year long altimetry record.

*corrected*

Rows 79-80

I suggest to eliminate this sentence, it is quite redundant.

*corrected*

Rows 81-87

In this paper we focus on the uncertainty in the GMSL record arising from instrumental errors in the satellite altimetry. The uncertainties of the measurements have been quantified in the GMSL record. This is an important information for the studies in the detection and attribution of the climatic changes, estimating the rise of global mean sea level as a response to the anthropogenic activity. In the detection-attribution studies the response of the GMSL to the anthropogenic activity needs to be separated from that one to climatic variability, representing an additional source of uncertainty.

*corrected*

Rows 87-89

I suggest to eliminate these two sentences. They are quite redundant.

*corrected*

Rows 98-101

We used an error budget approach to a global scale on a 10 day basis in order to estimate the error variance-covariance matrix. We considered all the major sources of uncertainty in the altimetry measurements, including the wet tropospheric correction, the orbital solutions and the inter-calibration of satellites. We have also taken into account the time correlation between the different sources of uncertainty (section 2). The errors have been separately characterized for each altimetry mission, since they have been affected by different sources of errors (section 2).

*corrected*

Rows 105-106

I suggest to eliminate this sentence, it is also very redundant.

1. GMSL data series

*corrected*

Rows 110-117

Each group processes the 1-Hz data with geophysical corrections to correct the altimetry measurement for various aliasing, biases and drifts, caused by different

atmospheric conditions, sea states, ocean tides and other causes (Ablain et al., 2009). They spatially average the data over each 10-day orbital cycle to provide GMSL estimates on a 10-day basis. The differences among the GMSL estimates from several groups arise from data editing, from difference in the geophysical corrections and from differences in the used method to spatially average the individual measurements during the orbital cycles (Masters et al., 2012; Henry et al., 2014).

*corrected*

Rows 117-121

Recently, the comparisons of the GMSL time series derived from satellite altimetry with independent estimates are based on the tide gauge records (Valledeau et al., 2012; Watson et al., 2015) or on the combination of the contribution to the sea level from thermal expansion, land ice melt and land-water storage (Dieng et al., 2017). They have shown that there was a drift in the GMSL record over the period 1993-1998. This drift is caused by an erroneous onboard calibration correction on TOPEX altimeter side-A (noted TOPEX-A).

*corrected*

2. Altimetry GMSL error budget

*corrected*

Rows 138-140

This section describes the different errors that affect the altimetry GMSL record. It builds on the GMSL error budget presented in Ablain et al. (2009) and extends this work taking into account the new altimeter missions (Jason-2, Jason-3) and the recent findings on altimetry error estimates.

*corrected*

Row 144

….. and by the correlation time-scale ($\lambda$).

*corrected*

Row 147

Add a point at the end of the sentence.

*corrected*

Row 149

The biases can arise ..

*corrected*

Row 159

The drifts may occur in the GMSL record….

*corrected*

Rows 163-170

This drift has been corrected by using several empirical approaches (Ablain, 2017; Beckley et al., 2017; Dieng et al., 2017), that are all affected by a significant uncertainty. We estimated this uncertainty to be….. with a comparison between an independent GMSL estimate based on tide gauge records (Ablain, 2017). For the TOPEX-B record, no GMSL drift has been reported, but Ablain et al. (2012) showed significant sigma-0 instabilities in the order of 0.1 dB, which generate through the sea-state bias correction an uncertainty…..(February 1999 – April 2002). Concerning the ITRF realization Couhert et al. (2015) have shown that…….

*corrected*

Rows 176-183

The residual time correlated errors are separated into two different groups, depending on their correlation time scales. The first group gathers errors with short correlation time scales, i.e. lower than two months and between two months and one year. The second group gathers errors with long correlation time scales between 5 and 10 years. In the first group the errors are mainly due to the geophysical corrections (ocean tides, atmospheric corrections), to the altimeter corrections (sea-state bias correction, altimeter ionospheric corrections), to the orbital calculation and to the potential altimeter instabilities (altimeter range and sigma-0 instabilities). At time scales below one year, the variability of the corrections' time series is dominated by errors, such that the variance of the error in each …..

*corrected*

Rows 184-186

For errors with correlation time scales lower than 2 months, we estimated the standard deviation ($\sigma$) of the error from the correction's time series filtered with a 2-month high-pass filter. Since the standard deviation of the errors depends on the different altimeter missions, the standard deviation has been separately estimated for each altimeter mission.

*corrected*

Rows 196-204

In the second group of residual time correlated errors, the errors are due to the onboard microwave radiometer calibration, yielding instabilities in the wet troposphere correction and also to the orbital calculation (Couhert et al., 2015). Since these errors are correlated at a time scale longer than 5 years, they can not be estimated with the standard deviation of the correction time series, too short (25-year long) to sample the time correlation. For this group of residual-time-correlated errors we used simple models to represent the time correlation of the errors. For the wet troposphere correction, several studies (Legeais et al., 2014; Thao et al., 2014; Fernandes et al., 2015) have identified long-term differences among the computed corrections from the different microwave radiometers and from different atmospheric re-analyses (Dee et al., 2011).

*corrected*

3. The GMSL error variance-covariance matrix

*corrected*

Rows 221-222

In this section we derived the error variance-covariance matrix ($\Sigma$) of the GMSL from the error budget described in the section 2. We assumed that all the error sources shown in Table 1 are independent one to each other.

*corrected*

Row 228

For the bias……

*corrected*

Row 231

alongtime

*corrected*

Row 234

….but not in the mean time GMSL

*corrected*

Row 236

This is the reason because
*corrected*
Row 238
For the drifts .....takes the shape
*corrected*

4. GMSL uncertainty envelope
*corrected*
Row 258
We estimated........
*corrected*
Rows 270-275
In Figure 4 we superimposed the GMSL time series (average of the GMSL time series in Figure 1) and the associated uncertainty envelop. For the TOPEX-A period we tested three different curves with three different corrections based on the removal of the Cal-1 mode (Beckley et al., 2017), on the comparison with tide gauges (Watson et al., 2015; Ablain, 2017) or based on a sea level closure budget approach (Dieng et al., 2017). The uncertainty envelop is centered on the record corrected for TOPEX-A drift with the correction based on Ablain et al. (2017). As it has been expected, all the empirically corrected GMSL records are within the uncertainty envelop.
5. Uncertainty in GMSL trend and acceleration
*corrected*
Rows 279-281
The variance-covariance matrix can be used to derive the uncertainty on any metric based on the GMSL time series. In this section we used the error variance-covariance matrix to estimate the uncertainty on the GMSL trend and acceleration over any period of 5 years and more within 1993-2017.
*corrected*
Rows 282-287
Recently, several studies (Watson et al., 2015; Dieng et al., 2017; Nerem et al., 2018; WCRP Global Sea Level Budget Group, 2018) have found a significant acceleration in the GMSL record from satellite altimetry (after correction for the TOPEX-A drift). The occurrence of an acceleration in the record should not change the estimation of the trend when calculated with a least squared approach. However, it can affect the estimation of the uncertainty on the trend. To cope with this issue, we address here at the same time both the estimation of the trend and acceleration in the GMSL record. In order to obtain this objective we used a second order polynomial as a predictor.
*corrected*
Rows 300-304
The most common method to estimate the GMSL trend and acceleration is the Ordinary Least Squares (OLS) estimator in its classical form (Cazenave and Llovel, 2010; Masters et al., 2012; Dieng et al., 2015; Nerem et al., 2018). This is also the most common method to estimate trends and acceleration in other climate essential variables (Hartmann et al., 2014 and references therein).
*corrected*
Rows 311-314
To address this issue, we used a more general formalism to integrate the GMSL error in the trend uncertainty estimation, following Ablain et al. (2009), Ribes et al. (2016) and IPCC AR5 (Hartmann et al., 2014; see in particular Box 2.2 and Supplementary Material).

*corrected*

Row 327

Eliminate the space

*corrected*

Rows 341-344 and Rows 344-346

Check the English form

*corrected*

Rows 354-355

The periods for which the acceleration in sea level is significant at the 90% confidence level are shown in Fig. 8.

*corrected*

Rows 362-363

It is unclear which is the relationship between the acceleration of Global Mean Sea Level and the volcanic eruptions (Mount Pinatubo).

*Church et al. 2005 showed that the impact of large volcanic eruptions on global ocean heat content is characterized by a rapid reduction in global ocean heat content during the year following the eruption followed by a period of recovery of a few years when global ocean heat content increases faster than before the eruption ( see also Gregory et al. 2006 and Delworth et al. 2005). The sez level record starts in oct 1992 which is 1.5 years after the eruption of Mount Pinatubo (15th of June 1991). At that time the global ocean heat content was starting to recover with an increasing rate of rise (see Fasullo et al. 2016, their fig.2) leading to an acceleration in sea level. We now explain this in the text ion line 611*

Rows 364-371

The period for which the trend in sea level is significant at the 90% confidence level is shown in Fig. 9. In periods when the acceleration is not significant, the second order polynomial that we used as a predictor to estimate the trend and the acceleration does not hold anymore in principle. For these periods we should turn out a first order polynomial. The use of a first order polynomial does not affect the trend estimates, but only the trend uncertainty estimates. We checked for differences in the trend uncertainty when using either second order or first order polynomial predictors. We found that these differences are negligible (not shown). Fig. 9 indicates that for periods of 5 years and longer, the trend in GMSL is always significant at 90% CL over the whole record. At the end of the record the trend tends to increase. This is consistent with the acceleration plot in Figure 6.

*corrected*

6. Conclusions

Row 379

......measurement also increases and the description of the errors improves.

*corrected*

Rows 383-385

The uncertainty of the GMSL here computed shows the reliability of altimetry measurements in order to accurately describe the evolution of the GMSL on all time scales from 10 days to 25 years. It also shows the reliability of altimetry measurements in order to estimate the trends and the accelerations of the sea level.

*corrected*

Row 387

…….we estimated……
*corrected*
Rows 391-394
In this study several assumptions have been made, that could be improved in the future. Firstly, the modeling of altimeter errors should be regularly revisited and improved to take into account a better knowledge of errors………

Concealing the mathematical formalism, the OLS method……..

CAPTIONS TO FIGURES

(from January 1993 to December 2017). I suggest to correct in all the captions.

Figure 1: Evolution of the GMSL series (corrected for TOPEX-A drift by using Ablain et al., 2009 TOPEX-A correction) from six different groups (AVISO/CNES, CSIRO, University of Colorado, SL_cci/ESA, NASA/GSFC, NOOA). The SL_cci/ESA covers a period from January 1993 to December 2016, while all the other products cover the full 25-year period (from January 1993 to December 2018). Seasonal (annual and semi-annual) signals have been removed and a 6-month smoothing has been applied. An averaged solution has been computed from the six groups. GMSL time series have the same average on the 1993-2015 period (common period) and the averaged solution starts at zero in 1993. The averaged solution without TOPEX-A correction has also been represented. A GIA correction of 0.3 mm/year has been subtracted to each dataset.

Figure 2: Error variance-covariance matrix of altimeter GMSL on the 25-years period (from January 1993 to December 2017).

Figure 3: Evolution in time of GMSL measurement uncertainty within a 90% confidence level (1.65 σ) on the 25-years period (from January 1993 to December 2017).

Figure 4: Evolution of the AVISO/GMSL with different TOPEX-A corrections. On the black, red and green curves, the TOPEX-A drift correction has been respectively applied based on Ablain (2017), Watson et al. (2015), Dieng et al. (2017) and Beckley et al. (2017). The uncertainty envelope, as well as the trend and acceleration uncertainties are given to a 90% confidence level (1.65 σ). Seasonal (annual and semi-annual) signals removed and 6-month smoothing have been applied. GIA correction has also been applied.

Figure 5: GMSL trend uncertainties (mm/yr) estimated for all altimeter period within a 25-years period (from January 1993 to December 2017). The confidence level is 90 % (1.65 σ). Each colored pixel respectively represents the half-size of the 90 % confidence range in the GMSL trend. The values are given in mm/y. The vertical axis indicates the length of the period (ranging from 1 to 25 years) considered in the computation of the trend, while the horizontal axis indicates the center date of the period (for example 2000 for the 20-year period 1990-2009).

Figure 8: GMSL acceleration using the AVISO GMSL time series corrected for the TOPEX-A drift using the correction proposed by Ablain (2017): the acceleration in the shaded areas is not significant (lower than the acceleration uncertainties at the 90% confidence level). The length of the window (in years) is represented on the vertical axis and the central date of the used window is represented on the horizontal axis.

Technical corrections
*corrected*
Row 288
… no observations
*corrected*
Row 455
SEA LEVEL BUDGET. Why capitals? Check and correct.

*corrected*
Row 478
Marine Geodesy, 35 (suppl. 1), 20-41
*corrected*
Row 503
Marine Geodesy, 35 (suppl. 1), 42-60.
*corrected*

---

## Author Response (AR2)

Comments to the Author (answer in blue):

1) There is an important missing point on page 5 line 141 (Erreur ! Source du renvoi introuvable).

=> Corrected

2) Non-public comments to the Author: Very few corrections due to the inclusion of new references and links to tables/figures. The references included in the revised text are underlined (word process consequence).

=> Corrected

[revised manuscript text omitted]